# AUX1-mediated root hair auxin influx governs SCF$^{TIR1/AFB}$-type Ca$^{2+}$ signaling

Julian Dindas[1], Sönke Scherzer[1], M. Rob G. Roelfsema[1], Katharina von Meyer[1], Heike M. Müller[1], K.A.S. Al-Rasheid[2], Klaus Palme[3], Petra Dietrich [iD] [4], Dirk Becker[1], Malcolm J. Bennett [iD] [5] & Rainer Hedrich[1]

Auxin is a key regulator of plant growth and development, but the causal relationship between hormone transport and root responses remains unresolved. Here we describe auxin uptake, together with early steps in signaling, in *Arabidopsis* root hairs. Using intracellular microelectrodes we show membrane depolarization, in response to IAA in a concentration- and pH-dependent manner. This depolarization is strongly impaired in *aux1* mutants, indicating that AUX1 is the major transporter for auxin uptake in root hairs. Local intracellular auxin application triggers Ca$^{2+}$ signals that propagate as long-distance waves between root cells and modulate their auxin responses. AUX1-mediated IAA transport, as well as IAA$^-$ triggered calcium signals, are blocked by treatment with the SCF$^{TIR1/AFB}$ - inhibitor auxinole. Further, they are strongly reduced in the *tir1afb2afb3* and the *cngc14* mutant. Our study reveals that the AUX1 transporter, the SCF$^{TIR1/AFB}$ receptor and the CNGC14 Ca$^{2+}$ channel, mediate fast auxin signaling in roots.

[1] Institute for Molecular Plant Physiology and Biophysics, University of Würzburg, 97082 Würzburg, Germany. [2] College of Science, King Saud University, Riyadh 11451, Saudi Arabia. [3] Institute of Biology II/Molecular Plant Physiology, Faculty of Biology, BIOSS Centre for Biological Signaling Studies, Centre for Biological Systems Analysis, 79104 Freiburg, Germany. [4] Molecular Plant Physiology, Department of Biology, University of Erlangen-Nürnberg, Staudtstrasse 5, 91058 Erlangen, Germany. [5] Centre for Plant Integrative Biology, Plant & Crop Sciences, School of Biosciences, University of Nottingham, Nottingham LE12 3RD, UK. Correspondence and requests for materials should be addressed to R.H. (email: hedrich@botanik.uni-wuerzburg.de)

Among all phytohormones auxin is unique for exhibiting polar transport[1]. The major form of auxin in higher plants, indole-3-acetic acid (IAA), is primarily synthesized in developing aerial tissues such as the shoot apex[2,3]. Early studies documented that auxin is mobilized, from its site of synthesis in the shoot apex, toward the root, by polar transport[4,5]. In the root, auxin accumulates at the tip and is then transported shoot-wards via the lateral root cap and epidermis employing a 'reverse fountain' transport mechanism[6,7].

The polar transport of auxin, from cell to cell, is mediated by specialized influx and efflux carriers[8–10]. PIN-FORMED (PIN) and P-GLYCOPROTEIN (PGP) gene family members encode the major auxin efflux carriers, whereas the AUXIN1/LIKE-AUX1 (AUX/LAX) gene family encodes the major auxin influx carriers[8,11]. Depolarization of the cell membrane represents one of the earliest responses evoked by IAA, which was first recognized in oat coleoptiles and root hairs of corn[12,13]. However, the molecular mechanisms underlying auxin-induced early membrane responses have remained elusive.

To study early steps in auxin signaling, we selected root hairs as our single-cell model. We employed Arabidopsis mutants defective for auxin transport and responses, in combination with membrane potential-, ion flux-, and calcium measurements. We report that early auxin-induced transient depolarizations represent AUX1-mediated uptake. We show that both the membrane electrical signal and a traveling $Ca^{2+}$ wave require $SCF^{TIR1/AFB}$ receptor-based activation of plasma membrane calcium channels.

## Results

**Auxin depolarizes root hairs in a dose-dependent manner.** Changes in membrane potential are often the earliest responses of plant cells to biological signals[14]. The membrane potential of a cell reflects the ensemble activity of ion channels, pumps, and electrogenic carriers. To monitor the membrane potential response to auxin, we used bulging root-hair cells of 3- to 5-day-old A. thaliana seedlings, grown on agarose plates. Under microscopic inspection, single-barreled microelectrodes were impaled at the tip of the root hairs (Fig. 1a), which assures that the tip of the electrode gets located in the cytosol[15]. With electrodes filled with 0.3 M KCl and a bath solution containing 1 mM $CaCl_2$; 0.1 mM KCl and 5 mM MES/BTP, pH 5.5 we measured an average plasma membrane potential of $-161$ mV $\pm$ 1 mV. To challenge the root-hair with auxin, we positioned a pressure-operated application pipette at a distance of ~150 μm (Fig. 1a) and applied back pressure during 1 s steps. A range of apparent IAA concentrations from 0.01 to 10 μM IAA was used (see Methods). Peak depolarizations of up to 70 mV with 10 μM auxin in the pipette were recorded (Fig. 1b). The apparent IAA concentration required for a half-maximal depolarization was 0.30 μM. The same value was found for the half-maximal velocity of the initial depolarization triggered by IAA (Fig. 1c). Hence, following auxin treatment, the membrane potential rapidly depolarized in a dose-dependent manner.

**AUX1 mediates proton-coupled auxin transport in root hairs.** Given that the auxin influx carrier AUX1 belongs to the amino acid permease family of proton-driven transporters[16,17] and IAA is derived from the amino acid tryptophan, we tested whether proton currents are accompanying IAA-triggered depolarization. Using pH sensitive extracellular electrodes that scan between two positions near the root hairs[18], we monitored proton efflux in the absence of IAA, which indicates that the membrane potential is dominated by the activity of the plasma membrane proton pump (Fig. 1d). Upon IAA application, the $H^+$-flux reversed its direction and an influx of $H^+$ was measured. Consistent with IAA triggering $H^+$-uptake because of the symport of IAA with protons into root hairs, we measured no, or only a minor, IAA-evoked depolarization at weakly alkaline and neutral pH, whereas the magnitude and velocity of the depolarization increased at more acidic pH values (Fig. 1e). The proton concentration required for a half-maximal IAA-dependent depolarization was 0.91 μM (pH 6; Fig. 1f). These data indicate that auxin and protons cross the cell membrane via $H^+$/IAA symport driven by the transmembrane voltage- and proton gradient.

Next, we quantified the $H^+$/IAA symport membrane phenomenon in response to IAA in root hairs of wild-type, versus a selection of auxin influx carrier (aux1) mutant alleles (Supplementary Fig. 1). All aux1 mutants exhibited a strongly reduced IAA-dependent depolarization (Fig. 2a). The partial loss-of function allele aux1-2 reduced IAA-induced depolarization by 50%, whereas the loss-of function allele aux1-T exhibited an up to 80% decrease (Fig. 2a). Proton flux recordings also revealed that aux1 mutants exhibited decreased $H^+$/IAA responses, with null alleles wav5-33 and aux1-T essentially lacking an IAA-dependent $H^+$-influx (Fig. 2a). Comparing $H^+$/IAA responses between wild-type and wav5-33 revealed that AUX1 contributes more than 80% to the total root-hair IAA uptake capacity (Fig. 2b; Supplementary Fig. 2a). AUX1 has a strong preference for naturally occurring IAA as substrate, based on a comparison of the root-hair depolarization of wild-type and wav5-33 mutants in response to the auxin analogs IAA, 5F-IAA, 1-NAA, 2-NAA, 2,4-D, and BA (Supplementary Fig. 2b). Hence, AUX1 is responsible for the proton-coupled IAA uptake into root hairs.

**AUX1-driven root-hair depolarization is sensitive to $[P]_{ext}$.** AUX1 is expressed in root epidermal atrichoblasts[19] and, to a lesser extent, in root hairs (trichoblasts)[20,21]. On the basis of membrane potential recordings we observed higher auxin transport activity in atrichoblasts compared to root hairs (Supplementary Fig. 3a), suggesting that higher transcript numbers translate into more active AUX1 transport protein. AUX1 expression and abundance is further induced during germination, as well as phosphate (P) starvation[22]. We therefore assumed that low phosphate supply would feedback on auxin transport capacity and performed membrane potential recordings on root hairs of Arabidopsis seedlings grown on phosphate concentrations ranging from 300 μM (standard MS medium) down to 0.3 μM (equivalent to low soil P levels). Resting membrane potentials were similar for both wild-type (163.1 mV $\pm$ 1.5 mV) and wav5-33 (163.8 mV $\pm$ 1.8) mutants irrespective of phosphate supply (Supplementary Fig. 3b). However, in wild-type root hairs fast membrane potential depolarization in response to a 1 s IAA pulse (0.3 μM) increased with decreasing phosphate supply and highest depolarization rates were obtained with seedlings grown in the presence of 0.3 μM phosphate (Fig. 2c, d). In contrast, the phosphate effect on membrane depolarization was absent in wav5-33 mutants (Fig. 2c, d). The observation that the IAA-induced instantaneous proton fluxes were higher at low phosphate conditions, suggests that this reflects differences in AUX1 activity (Supplementary Fig. 3c). Hence, our measurements reveal that AUX1 provides a key component of the auxin-dependent root-hair response to low external P.

**Auxin triggers elevation of cytosolic calcium in root hairs.** When root hairs were exposed to IAA pulses of 1 s duration, membrane depolarizations were evoked at pipette concentrations as low as 10 nM. Increased auxin concentrations enhanced the velocity, as well as the amplitude of the membrane potential changes (Fig. 1b, c). At IAA doses above 50 nM, the membrane potential changes were transient in nature. Such transients consist

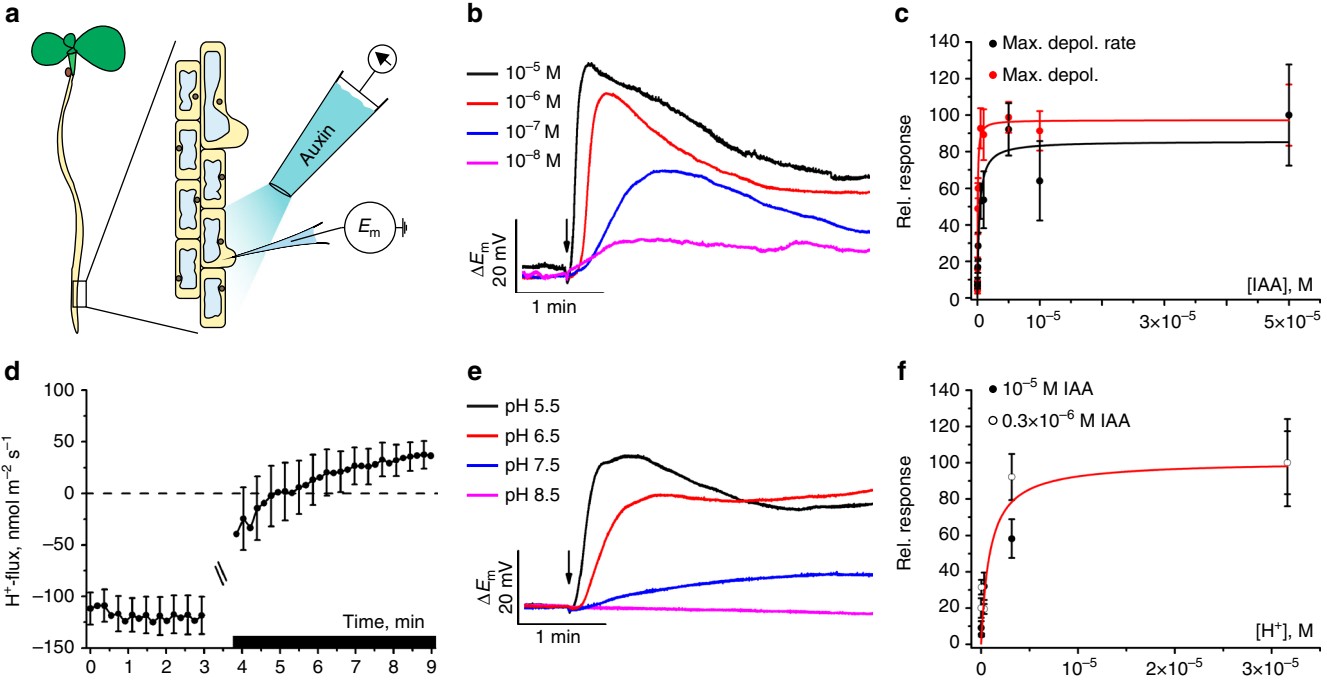

**Fig. 1** Auxin-induced plasma membrane depolarization and associated $H^+$-fluxes in root-hair cells. **a** Experimental set-up: Intracellular microelectrodes measured the membrane potential of bulging root-hair cells and auxin was applied with an application pipette by back pressure. **b** Representative changes in membrane potential evoked by 1 s IAA pulses ($10^{-8}$ to $10^{-5}$ M, arrow indicates time point of stimulation). **c** Dose–response curves of plasma membrane depolarization at a range of IAA concentrations, given as the percentage of the maximal depolarization (red symbols) and maximum rate of voltage change (black symbols). Apparent $K_m$ values according to Michaelis–Menten fitting were $53 \pm 6$ nM (max depol) and $300 \pm 133$ nM (max velocity) IAA ($n = 6 \pm$ s. e.m.). Note that $K_m$ values were calculated from undiluted auxin concentrations inside application pipettes. **d** $H^+$-flux kinetics determined with ion-selective electrodes, scanning in close proximity of root-hair cells. After 3 min, IAA was applied to a final concentration of 10 μM (black bar, curves are interrupted at time point of IAA stimulation), $n = 11 \pm$ s.e.m. **e** Impact of extracellular pH on the IAA-induced depolarization (arrow, 1 s stimulation with 10 μM IAA). **f** IAA-induced depolarization as a function of extracellular $H^+$-concentration, given as percentage of the depolarization rate at pH 4.5. Similar values were obtained for an IAA conc. of 10 μM (closed symbols) and 0.3 μM (open symbols). Apparent $K_m$ value according to Michaelis–Menten fitting was $910 \pm 500$ nM (i.e., pH 6.0; $n = 6$ for 10 μM IAA and $n = 10$ for 0.3 μM IAA). Error bars represent s.e.m.

of five phases: (i) an initial delay that precedes a (ii) period of acceleration followed by a phase in which (iii) the peak value is reached and (iv) a repolarization phase leading to (v) a steady state. Although the hormone was applied at the external root-hair side for only 1 s, the depolarization transient lasted for minutes and the plateau phase (v) even longer. This behavior points to an auxin response more complex than just AUX1-mediated $H^+$/IAA co-transport. Transient depolarizations followed by a prolonged depolarized phase have also been observed in response to the phytohormone ABA as well as microbe- or danger associated molecular pattern's (MAMPs, DAMPs)[23–26]. With the latter stimuli, depolarization increases cytosolic calcium concentration, and is likely to activate plasma membrane anion channels leading to an enhanced membrane depolarization[27,28].

To test whether IAA triggers a calcium influx, we performed measurements with scanning $Ca^{2+}$-selective extracellular electrodes. IAA provoked a transient influx of $Ca^{2+}$, with similar kinetics as observed for the simultaneously measured $H^+$-fluxes (Fig. 3a). However, in contrast to $H^+$-fluxes, inward $Ca^{2+}$-fluxes were transient in nature. To test if auxin-induced transient calcium influx leads to elevation of cytosolic $Ca^{2+}$ levels ($[Ca^{2+}]_{cyt}$), we studied *Arabidopsis* seedlings expressing the genetically encoded calcium reporters R-GECO1 and YC3.6[29,30], of which R-GECO1 shows high $Ca^{2+}$-dependent changes in fluorescence intensity[30]. A 1 s pulse of 10 μM IAA caused a transient rise of $[Ca^{2+}]_{cyt}$ that matched the $Ca^{2+}$-influx measured with ion-selective electrodes (Fig. 3b–d). Simultaneous membrane potential recordings and $Ca^{2+}$ imaging revealed that the cytosolic

calcium level returned to its resting level after 4–5 min, whereas membrane depolarization continued beyond this period (Fig. 3c). This suggests that the rise in $[Ca^{2+}]_{cyt}$ is not directly linked to the prolonged IAA-dependent membrane depolarization. This is further supported by different affinities for IAA transport and signaling (Supplementary Fig. 4a, b). The kinetics of both processes however matched each other during the initial phase of the IAA response (Supplementary Fig. 4c) indicating that auxin transport activity and cytosolic calcium changes are tightly coupled. This finding is further supported by the observation that calcium influx is impaired in *aux1* mutants (Supplementary Fig. 4d) and the finding that the auxin analogs 2-NAA and BA failed to efficiently trigger both responses (Fig. 3e).

**Calcium signals depend on auxin perception and CNGC14.** When root hairs were pretreated with the potent $SCF^{TIR1/AFB}$ auxin receptor inhibitor auxinole[31], the IAA-triggered depolarization rate was reduced to 20% of control cells. Moreover, auxinole completely repressed the transient increase in $[Ca^{2+}]_{cyt}$ (Fig. 3f), despite of the fact that auxinole shifts the root-hair resting membrane potential towards depolarized values (Supplementary Fig. 4e, f). This indicated that the IAA activated $SCF^{TIR1/AF}$ receptor complex is required to trigger the calcium transient and feeds back on AUX1 transport activity. To provide genetic evidence for the role of $SCF^{TIR1/AFB}$ we studied AUX1 activity and $Ca^{2+}$ signaling in *tir1*, as well as the *tir1afb2afb3* triple mutant (Fig. 4; Supplementary Fig. 4g, h). Both the IAA-induced membrane potential depolarization, as well as calcium influx,

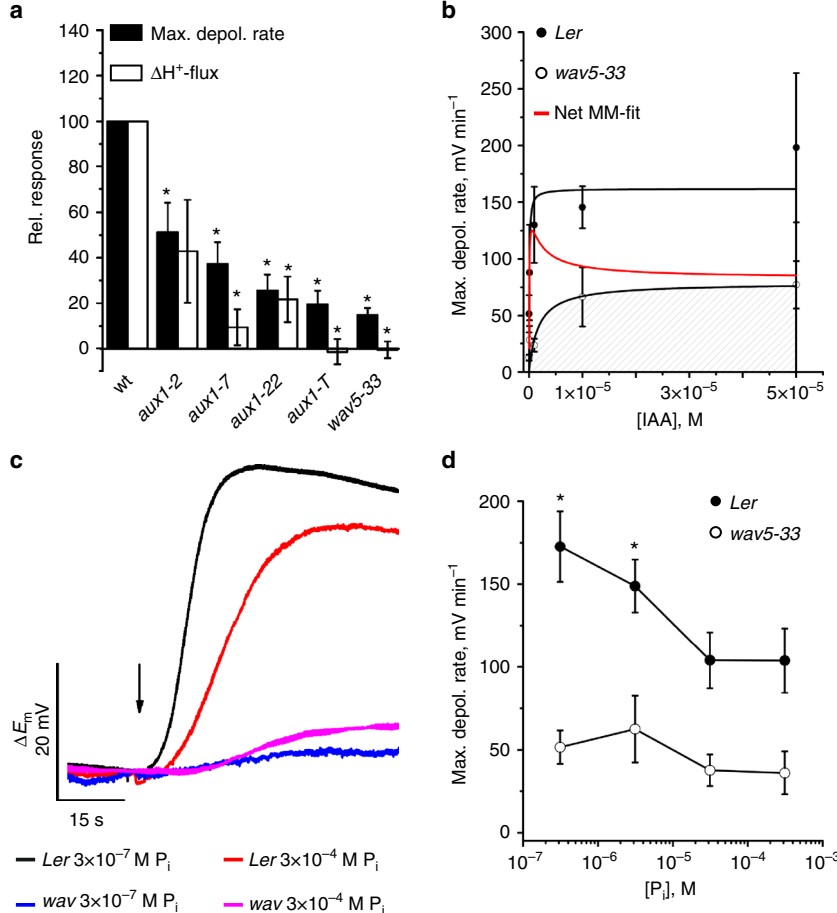

**Fig. 2** IAA-induced membrane depolarization and H$^+$-influx are AUX1-dependent. **a** Maximal depolarization rate and initial change of H$^+$-flux, evoked by 10 μM IAA in wild-type and *aux1-2* (*Ler*), *aux1-7* (*Col-0*), *aux1-22* (*Col-0*), *aux1-T* (*Ws*), and *wav5-33* (*Ler*). Values for mutants are given as the percentage of the response of the respective accessions (for mutants $n = 10 \pm$ SE for depolarization and $n = 12 \pm$ s.e.m. for H$^+$-flux, wild-type responses were measured with $n = 11$ to $13 \pm$ s.e.m.). Asterisks indicate significant reductions (Students *t*-test, $p < 0.05$). **b** IAA-induced max. depolarization rates, for *Ler* (closed symbols) and *wav5-33* (open symbols). Michaelis–Menten fits revealed half-maximal depolarization rates at $67 \pm 54$ nM for *Ler* and $1.7 \pm 1.6$ μM for *wav5-33*. Note that $K_m$ values were calculated from undiluted auxin concentrations inside application pipettes. Differential values of both black curves (red curve) point to a dominant AUX1-dependent uptake at [IAA] below $10^{-6}$ M, whereas unspecific transporters contribute at higher IAA-levels (*Ler* $n = 6$ to $8 \pm$ s.e. m.; *wav5-33* $n = 5 \pm$ s.e.m.). **c** IAA-dependent changes in the membrane potential of wild-type and *AUX1*-mutant, grown at standard and phosphate starved conditions. Representative traces of *A. thaliana Ler* and the *wav5-33* mutant following stimulation with 0.3 μM IAA during a period of 1 s (as indicated by arrow). **d** Dose–response curve of the phosphate concentration in the growth medium, plotted against the maximum rate of depolarization. Error bars show s.e.m. ($n = 14$ (*Ler*) and 9 (*wav5-33*)). Asterisks mark significant differences (Students *t*-test, $p < 0.05$)

were essentially unaffected in the *tir1* mutant, but were reduced to levels observed for auxinole treated root hairs in the triple mutant *tir1afb2afb3* (Fig. 4a, b). This suggests that the auxin receptor complex, including TIR1, AFB2, and/or AFB3, control AUX1 transport activity, as well as fast auxin signaling. Well in agreement with our hypothesis that functional auxin receptors are tightly linked to auxin transport and Ca$^{2+}$ signaling, the IAA-triggered H$^+$- and Ca$^{2+}$-influx were severely impaired in the *tir1afb2afb3* triple mutant when compared to wild-type, or *tir1* (Fig. 4b; Supplementary Fig. 4g, h). The transcript abundance for AUX1, as well as for the cyclic nucleotide-gated channel CNGC14 (see below), was unaffected in the receptor mutants, or by auxinole treatment (Supplementary Fig. 4i). This indicates that the SCF$^{TIR1/AFB}$ receptor modulates this response through a post-translational mechanism.

To test whether H$^+$/IAA import is directly coupled to calcium signaling (Fig. 3), we injected IAA into the root-hair cells directly. Doubled-barreled microelectrodes were placed in the root-hair cytoplasm, the membrane potential was monitored via the first barrel and on demand IAA could be iontophoretically injected via

the second microcapillary. Upon auxin injection the R-GECO signal reported an IAA-dependent Ca$^{2+}$ transient (Fig. 5a, b). The physiologically inactive 2-NAA failed to elicit the Ca$^{2+}$ response and in the presence of auxinole the Ca$^{2+}$ response was inhibited (Fig. 5c, d). The rise in cytoplasmic calcium evoked by IAA injected into the root-hair, however, was not superimposed by a fast membrane potential depolarization (Supplementary Fig. 5a). This shows that AUX1-driven H$^+$ import induces the fast depolarization, whereas cytosolic IAA only elicits the calcium signal, possibly via the SCF$^{TIR1/AFB}$ complex. Current injection of IAA caused a hyperpolarization of the plasma membrane of approximately $-10$ mV, followed by a slow depolarization of about 18 mV (Fig. 5c; Supplementary Fig. 5a). We hypothesized that the depolarization relates to calcium influx from the apoplast (Fig. 4a), possibly via the previously identified cyclic nucleotide-gated channel CNGC14[28]. Ion flux recordings and cytosolic auxin injection revealed that in line with cytoplasmic calcium imaging data[28], the IAA-triggered Ca$^{2+}$-influx was completely absent in the *cngc14* mutant (Fig. 5e; Supplementary Fig. 5b, c). Surprisingly, the IAA-induced membrane potential depolarization as

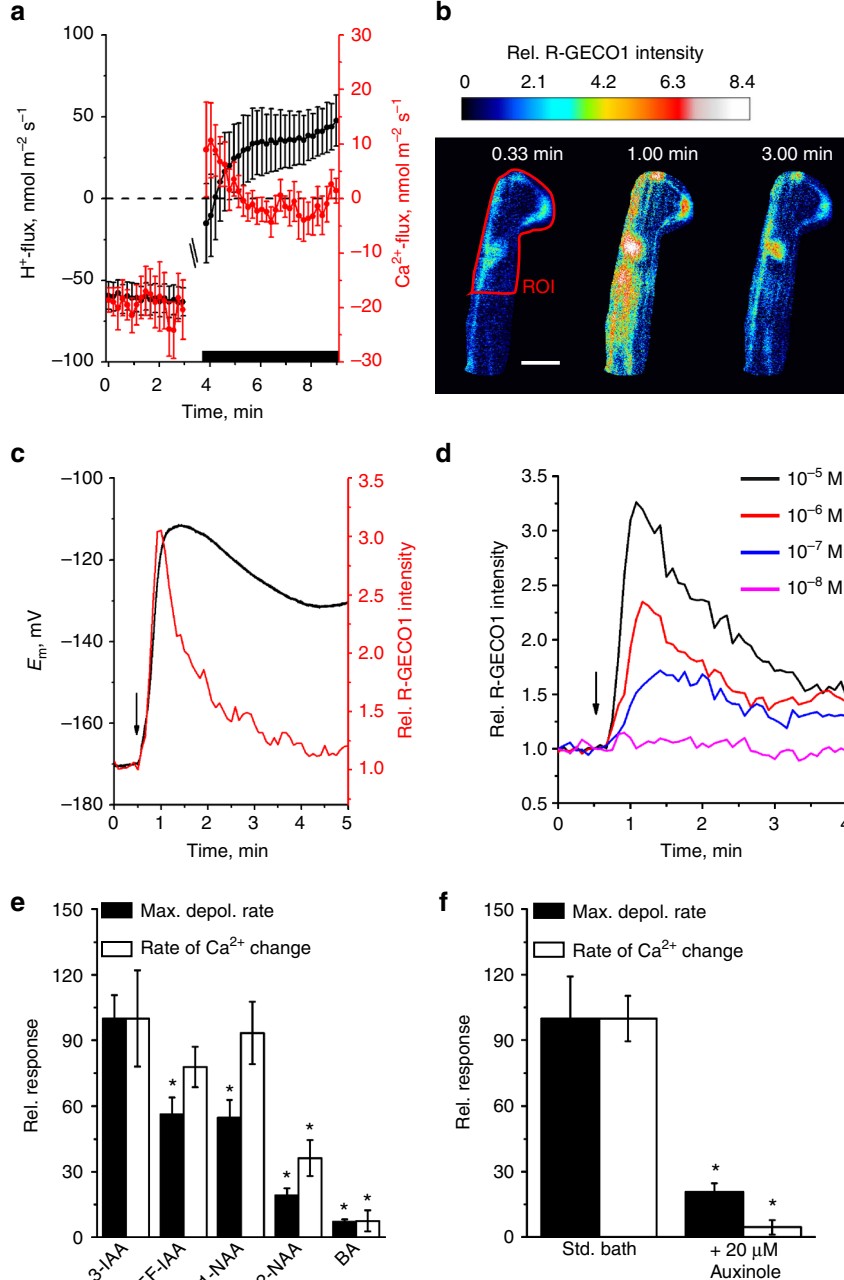

**Fig. 3** Auxin induces $Ca^{2+}$-influx and increase of the cytosolic-free $Ca^{2+}$-concentration of root-hair cells. **a** Average ion flux kinetics determined with ion-selective electrodes ($H^+$ black symbols, left axis; $Ca^{2+}$ red symbols, right axis, $n = 12 \pm$ s.e.m.). After 3 min, IAA was applied to a final concentration of 10 μM (black bar, curves are interrupted at time point of application). **b** IAA-evoked changes of the cytosolic-free $Ca^{2+}$-concentration in root-hair cells, determined with R-GECO1. Images of a single root-hair cell, before, during, and after the application of 10 μM IAA. IAA was applied at $t = 0.5$ min. False colored images indicate R-GECO1 fluorescence intensity, relative to the average value in ROI (red line) just before IAA application (color code above the panels). The scale bar represents 20 μm. **c** Simultaneous measurement of IAA-induced depolarization (black trace, left axes) and change in R-GECO1 fluorescence intensity (red trace, right axes) of a ROI that includes the nucleus, in a single root-hair cell (arrow: time point of 1 s application of 10 μM IAA). Representative measurement from 26 experiments. **d** IAA-dependent changes in R-GECO1 fluorescence intensity determined across the root and representative of 7 experiments for each IAA concentration (indicated by colored lines). **e** Max. depolarization rate (closed bars) and slope of R-GECO1 intensity change (open bars) of single root-hair cells to 1 s pulses of 10 μM IAA and auxin analogs, as indicated. Data are given as percentage of the response to 3-IAA ($n = 8 \pm$ s.e.m. for BA and $n = 16 \pm$ s.e.m. for auxins), asterisks indicate significant reductions (Students $t$-test, $p < 0.05$). **f** Max. depolarization rate (closed bars) and slope of R-GECO1 intensity change (open bars) of single root-hair cells to a 1 s pulses of 10 μM IAA. Root-hair cells were measured under control conditions, or after 20 min pretreatment with 20 μM auxinole. Bars represent percentage of control ($n = 10 \pm$ s.e.m. for control and $11 \pm$ s.e.m. for auxinole), asterisks indicate significant reductions (Students $t$-test, $p < 0.05$)

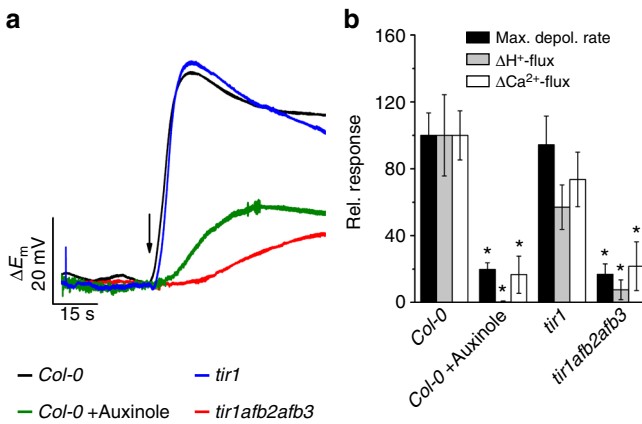

**Fig. 4** The auxin-induced plasma membrane depolarization as well as H⁺- and Ca²⁺-influx are dependent on intracellular auxin perception. **a** IAA-dependent changes in the membrane potential of wild-type and receptor mutant root hairs. Representative traces of *A. thaliana* Col-0 in the presence (green) and absence (black) of 20 μM auxinole as well as of the *tir1-1* (blue) and *tir1-1afb2-3afb3-4* (red) mutants following a 1 s stimulation with 10 μM IAA (arrow). **b** Maximal depolarization rate (black bars) and initial change of H⁺- (gray bars) and Ca²⁺-flux (open bars), evoked by 10 μM IAA in roots of *Col-0* ($n = 14 \pm$ s.e.m. for depolarization; $n = 10 \pm$ s.e.m. for ion fluxes), *Col-0* pretreated with 20 μM auxinole ($n = 6 \pm$ s.e.m. for depolarization; $n = 4 \pm$ s.e.m. for ion fluxes), as well as in *tir1-1* ($n = 8 \pm$ s.e.m. for depolarization; $n = 16 \pm$ s.e.m. for ion fluxes) and in *tir1-1afb2-3afb3-4* ($n = 6 \pm$ s.e.m. for depolarization; $n = 9 \pm$ s.e.m. for ion fluxes). Values are given as the percentage of the wild-type response. Asterisks indicate significant reductions (Students *t*-test, $p < 0.05$)

well as proton influx were absent in the *cngc14* mutant, too (Fig. 5e, f)[28]. Compared to wild-type, however, AUX1 as well as TIR1, AFB2, or AFB3 expression appeared unaltered in *cngc14* (Supplementary Fig. 5d). This indicates that CNGC14 regulates AUX1 activity at a post-transcriptional level. Further, this demonstrates that IAA is perceived by the SCF^TIR1/AFB auxin receptor complex and evokes a CNGC14-dependent Ca²⁺ signal. Transcriptional analyses revealed that auxin-inducible expression of IAA19 was strongly impaired in the *tir1afb2afb3* triple mutant, as well as in *cngc14* when compared to wild-type (Supplementary Fig. 5e). This observation is in line with an important role for CNGC14 in regulating AUX1-mediated auxin import.

**Auxin signaling in root cells is modulated by calcium waves.**
We observed that the elevation of $[Ca^{2+}]_{cyt}$ in root hairs not only occurred in those cells injected with IAA, but also in those that are symplastically connected to it (Fig. 5a, b). The calcium signal propagated as a transversal wave from the IAA stimulated root-hair towards the center of the root and beyond that to epidermal cells at the opposite side of the root. This Ca²⁺ wave traveled at a speed of 0.5 cm h⁻¹ and traversed the entire root profile within two minutes (Fig. 5b). The Ca²⁺ signal underwent amplification while spreading through the root, as secondary Ca²⁺ signals (ROI2) were about twofold higher when compared to those at the site of origin (ROI1, Fig. 5b, d). As seen for the local depolarization response, the inactive auxin 2-NAA failed to elicit Ca²⁺ waves, whereas the auxin response inhibitor auxinole blocked auxin-induced Ca²⁺ signaling (Fig. 5d).

Upon reaching the root stele, the Ca²⁺ wave propagated tipwards with a constant speed of 1.56 cm h⁻¹ (Fig. 6a, b; Supplementary Fig. 6a). Auxin transport also can occur at these velocities[32]. Because of the constant velocity of propagation however, the Ca²⁺ signals most likely resemble Ca²⁺ waves such as those that have been observed in a variety of eukaryotes[33–35].

Such a Ca²⁺ wave may modulate auxin signaling in cells, at distance from the IAA injected one. To challenge this hypothesis we took advantage of the Aux/IAA-based auxin-signaling sensor DII-VENUS[36]. Upon injection of IAA into a single root-hair cell, the DII-VENUS signal in cells of the elongation zone (ROI in Fig. 6c) started to decay within minutes, to reach a new steady state within 25 min (Fig. 6d). We investigated the possible causal relationship between calcium and auxin signaling by repeating these experiments in the presence of the Ca²⁺ channel blocker lanthanum. A concentration of 128 μM La³⁺ was chosen, as it entirely repressed a second IAA-dependent depolarization that occurs in root hairs exposed to two consecutive hormone pulses in the absence of IAA (Supplementary Fig. 6b). When La³⁺ was applied at this concentration, it inhibited both IAA-triggered Ca²⁺ signaling and DII-VENUS degradation at the root tip (Supplementary Fig. 6c; Fig. 6d). Hence, we conclude that AUX1-regulated local auxin signaling in root hairs triggers a Ca²⁺ wave that controls auxin action, both at short and long distances.

## Discussion

To study the very earliest auxin responses in *Arabidopsis* root hairs, we employed electrophysiological techniques. This revealed that IAA depolarizes the plasma membrane in an AUX1-dependent manner. On the basis of a stoichiometry of at least two H⁺ per IAA⁻ and a chemiosmotic potential ranging from 180 to 300 mV, our data indicate that AUX1-mediated transport could theoretically accumulate auxin to 10⁵ fold higher concentrations inside root hairs, as compared to the cell walls. The in planta affinity of AUX1-dependent IAA transport determined in this study (0.3 μM) is higher compared to those obtained in heterologous expression systems[37,38], but in close agreement with values found for auxin-dependent degradation of Aux/IAAs mediated by TIR1[39]. Consistent with this major role, mutants lacking AUX1 are profoundly impaired in root development and nutrient controlled root adaptive responses[40–42]. The observed difference in AUX1-dependent membrane depolarization under PO₄ limiting conditions, suggest AUX1 is important to auxin-dependent adaptive root responses. Phosphate-starved roots were also characterized by increased auxin levels and auxin responses[43,44]. This is related to the enhanced expression of the auxin receptor *TIR1*[44] and reveals coupling of auxin transport and perception, to fine-tune root auxin responses.

Auxin-regulated developmental processes are usually described to be dependent on transcriptional changes that occur over a timescale of several minutes to hours[45]. Our ion flux studies revealed that application of local auxin pulses virtually immediately result in inwardly directed proton fluxes in accordance with experiments employing genetically encoded pH sensors[46]. These proton fluxes are largely reduced in *aux1* mutants, consistent with the auxin-induced rise in extracellular pH being dominated by AUX1-mediated 2H⁺/IAA⁻ transport activity. Our studies reveal that IAA triggers cytoplasmic Ca²⁺ transients, which require SCF^TIR1/AFB-based auxin signaling. Auxinole suppresses the auxin-induced Ca²⁺ response almost completely and functional studies employing the *tir1afb2afb3* triple mutant suggest a prominent role for SCF^TIR1/AFB in fast auxin signaling (Fig. 4). In response to 1 s IAA pulses, the membrane potential depolarizes at the same velocity as the transient Ca²⁺ elevation is initiated (Supplementary Fig. 4c). Again, this can only be explained by a very short auxin-signaling pathway, in which IAA entry via AUX1 binds to SCF^TIR1/AFB, which in turn regulates opening of plasma membrane calcium channels. We found that loss-of-function mutants of the cyclic nucleotide-gated channel CNGC14 lack an IAA-triggered Ca²⁺, as well as an AUX1-mediated

$H^+$-influx. In line with previous studies[28,47], our findings support a model in which CNGC14 acts as the bona fide auxin-activated $Ca^{2+}$-permeable channel at the plasma membrane (Fig. 7). This fast, membrane-delimited auxin signaling mechanism is unlikely to require downstream components, such as nuclear localized AUX/IAA proteins that regulate gene transcription[48]. This situation is reminiscent of fast and slow ABA signaling events[49]. In analogy to fast ABA signaling, one would expect that the auxin-activated $SCF^{TIR1/AFB}$ complex regulates a protein phosphatase, or kinase, to open the CNGC14 channel. The impact of CNGC14 on $[Ca^{2+}]_{cyt}$ probably is crucial for AUX1 activity, since AUX1-mediated IAA transport is inhibited in the presence of the $Ca^{2+}$ channel blocker $La^{3+}$, as well as in cngc14 (Supplementary

Fig. 6, 7). This suggests a negative, $Ca^{2+}$-dependent, feedback control circuit within a functional signaling pathway consisting of AUX1, $SCF^{TIR1/AFB}$, and CNGC14.

Maintenance of a tip-focused cytoplasmic calcium gradient is essential for root-hair growth[50]. Consistent with such a model, the R-GECO reporter revealed an auxin-induced $Ca^{2+}$ signal at the root-hair tip, but this was transient (Fig. 6b). Hence, a prolonged and elevated auxin source (for example, under low external $PO_4$ conditions) may be necessary to maintain the tip-focused $Ca^{2+}$ gradient and promote hair elongation. During root-hair growth, the tip-focused $Ca^{2+}$ gradient is known to coordinate secretion, endocytosis and actin dynamics[51]. In contrast, the role of the observed auxin-inducible intercellular calcium wave is less clear. One possibility is that the calcium wave modulates auxin transport[45]. Consistent with such a model, we observed changes in the abundance of the auxin sensor DII-VENUS (Fig. 6). Blocking $Ca^{2+}$ signaling via $La^{3+}$ inhibits DII-VENUS degradation and thus distal auxin signaling (Fig. 6d). This calcium wave may therefore help to modulate auxin responses in surrounding root epidermal cells to coordinate hair elongation and improve foraging for $PO_4$. However, further studies are required to test these intriguing mechanistic possibilities. Nevertheless, the spatial and temporal resolution of our electrophysiological-based auxin transport measurements, combined with imaging dynamic changes in auxin and $Ca^{2+}$ abundance within and between root-hair cells, provides insight into the auxin-dependent fast signaling events taking place in epidermal cells, at the interface between the root system and nutrient sources in the soil.

## Methods

**Plant growth conditions.** Seeds of *Arabidopsis thaliana* were sterilized for 5 min by application of 6% NaOCl, supplemented with 0.05% Triton-X 100. Three to six washing steps with deionized water removed the sterilizing solution. Single seeds

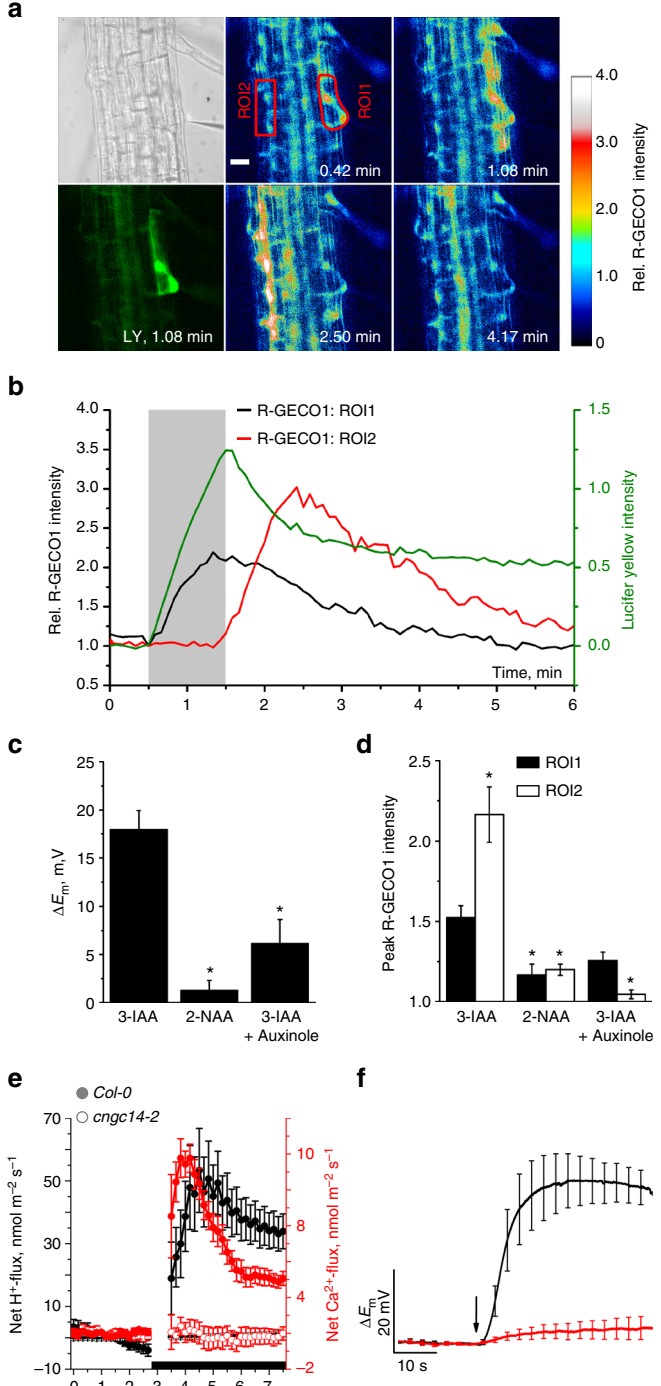

**Fig. 5** Cytosolic application of IAA triggers a lateral $Ca^{2+}$ wave in roots. **a** IAA was applied iontophoretically together with Lucifer yellow (LY) via an intracellular double-barreled microelectrode. Panel upper left, transmitted light signal, note microelectrode on right. Panel lower left, LY fluorescence signal at indicated time value. Panels middle and right, false color images of R-GECO1 intensity, normalized to the average value of ROI1 just before stimulation with IAA, as indicated by the scale on left. Time after experiment onset is given. IAA was injected with a current of -1 nA to the cell in ROI1, from $t = 0.5$ to 1.5 min. Representative measurement from 20 experiments. The scale bar represents 20 μm. **b** Representative time-dependent changes in the fluorescence signal of LY (green line, right axis) and the R-GECO1 signals of ROI1 (black line, left axis) and ROI2 (red line, left axis), relative to their average fluorescence intensity at the start of injection. The light gray bar indicates the period of iontophoretic injection of IAA. Representative measurement from 20 experiments. **c, d** Responses of *A. thaliana Col-0* root-hair cells to iontophoretical intracellular injection of IAA, 2-NAA, and IAA in roots pretreated with 20 μM auxinole. **c** Average value of max. depolarization. **d** Average changes in R-GECO1 signals of ROI1 (closed bars) and ROI2 (open bars, $n = 20 \pm$ s.e.m. for IAA; $n = 11 \pm$ s.e.m. for 2-NAA and $n = 6 \pm$ s.e.m. for IAA w/auxinole), asterisks indicate significant differences (Students t-test, $p < 0.05$). **e** Average net $Ca^{2+}$- (red) and $H^+$- (black) flux kinetics of wild-type (closed symbols) and cngc14-2 (open symbols) roots. After 3 min, IAA was applied to a final concentration of 10 μM (black bar), curves are interrupted at time point of IAA stimulation, $n = 17 \pm$ s.e.m. for Col-0 $Ca^{2+}$-fluxes, $n = 18 \pm$ s.e.m. for cngc14-2 $Ca^{2+}$-fluxes, $n = 7 \pm$ s.e.m. for $H^+$-fluxes. **f** IAA-dependent changes in the membrane potential of wild-type and cngc14-2 mutant root hairs. Average traces of *A. thaliana Col-0* (black) and of the cngc14-2 (red) mutant following a 1 s stimulation with 10 μM IAA (arrow) ($n = 6 \pm$ s.e.m. for Col-0 and $n = 7 \pm$ s.e.m. for cngc14-2)

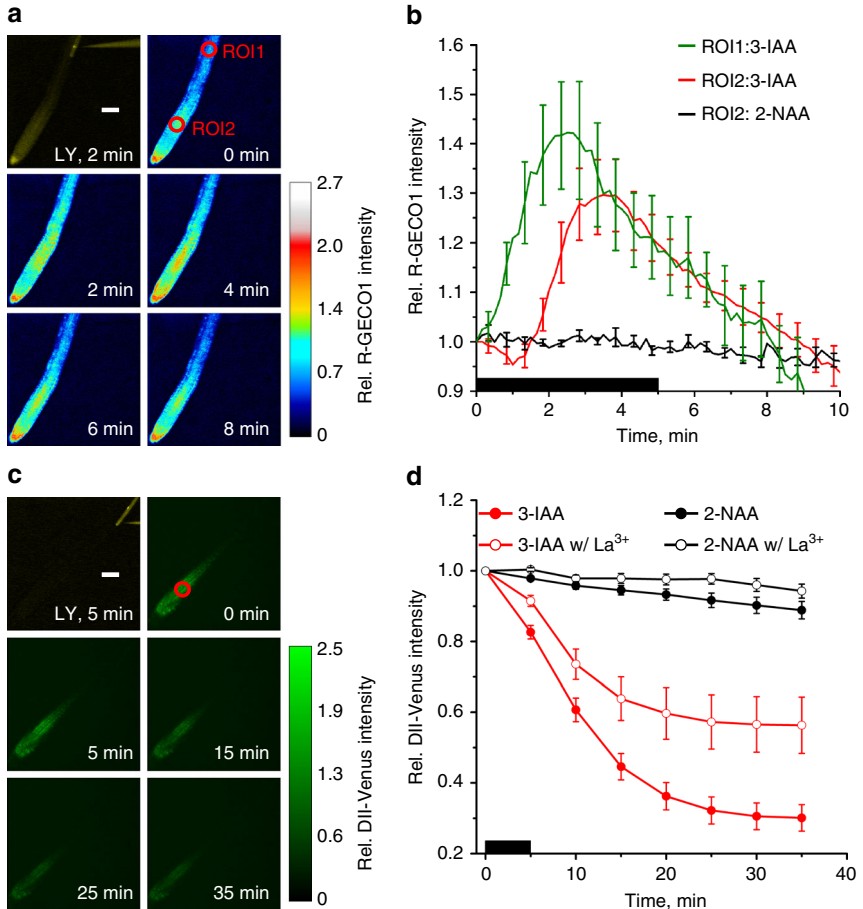

**Fig. 6** Long-distance $Ca^{2+}$- and IAA-signaling. **a** Fluorescence image of a single root-hair cell stimulated by intracellular application of IAA and LY, over a period of 5 min. All other panels, false color images indicating changes in R-GECO1 intensity, normalized to the average value at ROI2 at the start of IAA-injection (indicated by the color bar on the left). The time after the start of the experiment is given. Representative for 8 measurements. The scale bar represents 100 µm. **b** Changes in R-GECO1 signal intensity, plotted against time, in ROI1 (green line) and ROI2 (red line) of root stimulated with 3-IAA, or in ROI1 of root stimulated with 2-NAA (black line). Data are presented relative to the average fluorescence intensity in ROI of interest, at the start of injection. Black bar indicates the period of IAA injection into a root epidermal cell in close proximity to ROI1. Average values are shown ($n = 8 \pm$ s.e.m. for IAA and $n = 8 \pm$ s.e.m. for 2-NAA). **c** Fluorescence image of a single root-hair cell stimulated by 5 min intracellular application of IAA and LY. All other panels, changes in fluorescence intensity of the auxin-signaling sensor DII-Venus triggered by cytosolic application of IAA, bar on the right indicates fluorescence intensities relative to the average intensity of the ROI at the start of the experiment. Representative for 14 experiments. The scale bar represents 100 µm. **d** Changes in DII-Venus signal intensity, relative to the start of the experiment. Single root epidermal cells were stimulated with 2-NAA (black curves), or 3-IAA (red curves), in the presence (open circles) or absence (closed circles) of 128 µM $La^{3+}$ in the bath. Data were obtained from ROI at ~400 µm distance, towards the root tip, from the stimulated root epidermal cell (red circles in **a** and **c**). Average values are shown ($n = 14 \pm$ s.e.m. w/o $La^{3+}$ and $n = 6 \pm$ s.e.m. w/$La^{3+}$)

were placed in a row on the surface of small petri-dishes (Ø 35 mm) filled with 1 ml of plant growth medium (0.12% Murashige and Skoog basal salt mixture incl. MES, Duchefa; 0.5% sucrose; 1% agarose, pH 5.8 with TRIS). The petri-dishes were placed vertically in a growth chamber (KBWF 720, Binder) with controlled environmental conditions (12 h day vs. 12 h night; 21 °C at day vs. 16 °C at night; 120 µmol photons $m^{-2}$ $s^{-1}$) 3 to 5 days before experiments. For phosphate starvation experiments, ¼-strength MS-media (5.2 mM $NH_4NO_3$; 4.7 mM $KNO_3$; 0.6 mM $CaCl_2$; 0.2 mM $MgSO_4$; 25.1 µM $H_3BO_3$; 25 µM $Na_2EDTA$; 25 µM $FeSO_4$; 19 µM $MnSO_4$; 13.3 µM $ZnSO_4$; 1.7 µM KI; 386 nM $Na_2MoO_4$; 30 nM $CoCl_2$; 25 nM $CuSO_4$; 312 µM–0 µM KCl; 312 µM–0.3 µM $KH_2PO_4$; 0.5 % sucrose; 2.35 mM MES; pH 5.8 with TRIS) were manually prepared and plants were grown for 4 days on those media.

**Membrane potential recordings and auxin application.** Sterile grown seedlings were exposed to standard bath solution (0.1 mM KCl, 1 mM $CaCl_2$, 5 mM MES/BTP pH 5.5; in the case of $La^{3+}$-containing bath solutions ionic strength was kept constant and the sum of $La^{3+}$ and $Ca^{2+}$ concentrations was 1 mM). The response of the membrane potential of root epidermal cells to apoplastic and cytosolic auxin application was measured by sharp single- and double-barreled microelectrodes impaled through the tip of bulging root-hair cells. Fabrication of single- and double-barreled microelectrodes from borosilicate glass capillaries and membrane potential recordings were as described by Wang et al.[15] Back-pressure-operated

application pipettes for apoplastic auxin application were made of single-barreled microelectrodes, of which the tips were broken off, to obtain an opening of 20–40 µm. The application pipettes were filled with bath solution containing the required concentration of auxin and back-pressure pulses of 1 s were applied through a Picospritzer II microinjection dispense system (General Valve) operating at 30 psi. A range of IAA concentrations from 0.01 to 10 µM IAA was used, but the hormone concentration at the root cell plasma membrane will have been lower. The application pipette was operated at a distance of 150 µm of the root and experiments with the fluorescent dye Lucifer yellow CH (Fluka/Sigma) revealed an average fourfold dilution solutes in the pipette over the distance of 150 µm. The tip of single and double-barreled microelectrodes, used for iontophoretic injection of auxin into the cytosol, was filled with bath solution containing 6.66 mM 3-IAA (Sigma-Aldrich), 3.33 mM 2-NAA (Merck), 0.5 mM Lucifer yellow and 0.83 mM HEPES/TRIS (pH 7), whereas the remaining of the electrode was filled with 300 mM KCl. For cytosolic injection a current of −1 nA was applied.

**Ion flux measurements.** Ion fluxes across the plasma membrane of root epidermal cells were monitored using non-invasive ion-selective scanning microelectrodes. To avoid rupture of the root-hairs due to contact with the vibrating electrodes, the tip of the electrode was placed on the root body at the early differentiation zone between root-hairs. Prior to experiments, *Arabidopsis* seedlings were accustomed to the bath solution (100 µM KCl, 100 µM $CaCl_2$, 100 µM MES/BTP pH 5.5) for at

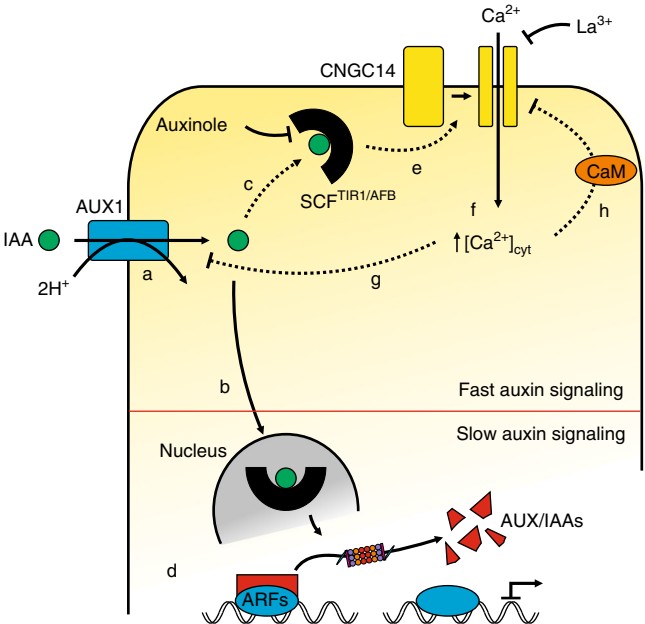

**Fig. 7** Model for fast auxin signaling in root cells. Following its uptake via the high-affinity IAA/$H^+$-symporter AUX1 (a) auxin is perceived by a nuclear (b) and a cytosolic fraction of the $SCF^{TIR1/AFB}$ receptor complex (c). Although nuclear perception results in changes at the transcriptional level via proteasomal degradation of Aux/IAA-class transcriptional repressors (d), a functional complex between the auxinole-sensitive cytosolic $SCF^{TIR1/AFB}$ receptor and CNGC14 activates $La^{3+}$-sensitive $Ca^{2+}$-influx upon auxin perception (e) resulting in elevated cytosolic $Ca^{2+}$ levels (f). Increased $[Ca^{2+}]_{cyt}$ feeds back on AUX1-mediated auxin transport, possibly through phosphorylation of AUX1 via $Ca^{2+}$-dependent kinases (g) and on CNGC14 activity via the $Ca^{2+}$-dependent interaction of CNGC14 with calmodulin (h)[52]

least 20 min. After stable fluxes were recorded for at least 3 min, 3-IAA was added to the bath solution to a final concentration of 10 μM. Measurements in which plants were treated with a solvent control (EtOH) showed no flux alterations (Supplementary Fig. 8). Electrode preparation, calibration and experimental set-up are described in the Supplementary Methods.

**Calcium and DII-Venus imaging**. Life cell imaging was performed on a Zeiss Axiokop 2FS, equipped with a CARV2 confocal imaging unit (Crest Optics). Image acquisition was carried out without the spinning disc in the light path, using a charge multiplying CCD camera (QuantEM 512SC, Photometrics), controlled by Visiview software (Visitron). The following single bandpass excitation filters were used; 430 nm (ET 430/24 nm, Chroma technology), 500 nm (filter 500/20, Semrock) and 562 nm (562/40 nm, Semrock) for LY, DII-Venus and R-GECO1, respectively. Light was reflected on the sample by a 490 nm dicroic mirror (Zeiss), a 444/521/606 dichroic beamsplitter (Brightline triple-edge beamsplitter, Semrock) and a 590 nm dichroic mirror (Zeiss). The emission signals were filtered with single bandpass filters; 520/30 nm (Semrock), 520 nm (Brightline HC 520/35, Semrock) and 628 nm (628/40 nm, Semrock). The excitation light was focused on the sample through an Achroplan ×40/0.80w objective (Zeiss) or an Achrostigmat ×10/0.25 objective (Zeiss). For analysis of imaging data the ImageJ Software (imagej.nih.gov/ij/) was used.

**Quantitative real-time PCR**. RNA was isolated from whole seedlings after electrophysiological experiments. To obtain an adequate amount of material for extraction approx. 5 to 10 seedlings were pooled into one sample. For RNA extraction the NucleoSpin® RNA Plant Kit (Macherey-Nagel) was used and transcript levels were ultimately analyzed after cDNA synthesis through quantitative real-time PCR using the Absolute QPCR SYBR green capillary mix (Thermo Scientific) and a Realplex Mastercycler (Eppendorf). Expression levels of individual genes were calibrated to standard curves for each transcript and subsequently normalized to 10,000 transcripts of actin (AtACT2/8). The following primers were used: AtACT2/8: AtACT2/8 forward (fw): 5′-GGTGATGGTGTGTCT-3′, AtACT2/8 reverse (rev): 5′-ACTGAGCACAATGTTAC-3′; AtAUX1: AtAUX1fw: 5′-GGATGGGCTAGTGTAAC-3′, AtAUX1rev: 5′-TGACTCGATCTCTCAAAG-3′; AtCNGC14: AtCNGC14fw: 5′-CAGCCAAGCTAAGACTCT-3′, AtCNGC14rev: 5′-GTTGAAGCCTTTGCTTTA-3′; AtTIR1: AtTIR1fw: 5′-

CTTCTTGTTCCGTGAGTT-3′, AtTIR1rev: 5′-ATTCAAATTATTGGCGAC-3′; AtAFB2: AtAFB2fw: 5′-ATGATAATAACCGGATGGA-3′, AtAFB2rev: 5′-TCGGGAAAGACACACTAAC-3′; AtAFB3: AtAFB3fw: 5′-GACGTGGGTAGG-TACGAAA-3′, AtAFB3rev: 5′-AAAACACATGAAGGTGCA A-3′; AtIAA19: AtIAA19fw: 5′-TTGATAAGCTCTTCGGTTT-3′, AtIAA19rev: 5′-TGCAAAA-TAATATACTATAAC-3′

**Statistical procedures**. To test significances two-tailed Student's $t$-tests were applied. All data points marked as significantly different from wild-type or control experiments were so with $p < 0.05$.

**Data availability**. The authors declare that all data supporting the findings of this study are available within the article and its Supplementary Information files, or from the corresponding author on request.

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

## Acknowledgements

This work was supported by the German Research Foundation (DFG, FOR964 to R.H. and D.B.). We thank Dr. Ken-Ichiro Hayashi for providing auxinole and Pamela Korte for qPCR assistance.

## Author contributions

R.H., M.J.B., D.B., P.D., K.P., K.A.S.A-R., M.R.G.R., and J.D. designed the study and wrote the manuscript. J.D., S.S., K.v.M. and H.M.M. performed the experiments and analyzed the data. M.R.G.R. wrote the LabVIEW-based software for analyzing ion flux data.
