## [Peer Review File(PDF 510 kb) · Nature Communications]

Reviewers' comments:

Reviewer #1 (Remarks to the Author):

This is an interesting paper that reports novel findings on the mechanisms of auxin uptake and the auxin-mediated early signaling steps in Arabidopsis root hairs. Perhaps not completely surprising given the short root hair phenotype of aux1q seedlings, the authors demonstrate that AUX1 is the major transporter of auxin in root hairs. The authors also report that auxin triggers a transient elevation in cytosolic Ca²⁺ in root hairs, and that Ca²⁺ signals propagate as long distance waves between root cells. It was also shown that treatment with an inhibitor of the auxin receptor TIR1 blocked this Ca signals.

As such the data described in this paper is novel and interesting. In particular, I find that the discovery that TIR1 is involved in very rapid signaling processes opens a new concept as to how protein degradation could be involved in processes that take few seconds to activate. I have some concerns that the authors should address before this manuscript can be accepted for publications:

1) To really confirm the role of TIR1 in mediating Ca waves, it would be necessary to examine a KO TIR1 mutant. Pharmacological treatment always raises the concern that they might affect other proteins in a non-specific manner. Thus, it cannot be ruled out that auxinole could be affecting components of the auxin signaling pathway other than TIR1.

2) The results reported on the effect of Pi on auxin transport into root hairs are not completely clear. If Pi signaling leads to an up-regulation of AUX1 expression, a higher level of this auxin transport is expected. Thus, it is not a very novel finding, unless it is proven that Pi signaling affects AUX1 transport efficiency or proton availability in a manner that is independent on the level of AUX1.

3) It is a bit contradictory that the depolarization experiments reported in this work, show that IAA has a strong preference for IAA than synthetic auxins, while in previous works it has been reported that NAA and 2,4 D have been a higher affinity for auxin uptake transporters than IAA, and that is why they are more active and often used in tissue culture. The authors should explain this discrepancy

Reviewer #2 (Remarks to the Author):

In this manuscript by Dindas et al, the authors investigate mechanisms underlying rapid auxin-induced ion signaling in Arabidopsis root hairs. Electrophysiological and fluorescence microscopy approaches are used to show that IAA treatment triggers rapid plasma membrane depolarization in root hairs and a concomitant increase in cytosolic Ca²⁺ levels that propagates into the root elongation zone. Based on their results, the authors propose the following model: Auxin-induced membrane depolarization is caused by co-transport of $\geq 2H^+$ /IAA⁻ mediated by the auxin influx carrier AUX1. Auxin taken up by root hairs then activates Ca²⁺ permeable channels via a signaling pathway that requires the auxin receptor(s) TIR1/AFB. Ca²⁺ signals subsequently propagate into the root elongation zone to modulate auxin signaling.

Identifying molecular components of non-transcriptional auxin signaling networks has been an important research area for decades, and some of the results described in this paper offer potentially novel and exciting insights. However, I think that several issues need to be addressed to strengthen some of the findings and conclusions.

(i) A key conclusion of this work is that TIR1/AFB auxin receptors are required to activate Ca²⁺ channels upon a rise in cytosolic auxin concentration. The discovery of a non-transcriptional signaling role for TIR1/AFB would be a major breakthrough. However, as far as I can tell, the authors deduce this solely on the strength of a pharmacological assay using the TIR1/AFB inhibitor auxinole. I don't understand how the authors can be quite so certain that auxinole is specific to TIR1/AFB? If there is

another, not yet identified auxin receptor, this receptor could clearly also be a target of auxinole. It was previously shown that auxin-induced, Ca²⁺-dependent ion signaling also occurs in the tir1afb2afb3 triple mutant (Monshausen et al 2011). How can this be reconciled with the conclusions described in this ms? Do root hairs and epidermal cells of the root elongation zone employ very different strategies to activate Ca²⁺ signaling? The authors need to provide better support for such a novel role of TIR1/AFB (e.g. mutant analyses, analysis of auxinole effects in apical root tissues).

(ii) It has previously been shown that AUX1 is not normally detectable in root hairs (Jones et al, 2009, Nat Cell Biol 11: 78–84). Unless I missed it, no evidence is provided in this or the accompanying manuscripts that phosphate starvation promotes AUX1 expression in root hairs. The authors must verify that such an increase in AUX1 levels occurs (e.g. using transgenic Arabidopsis expressing AUX1::AUX1-YFP - Jones et al 2009, or via qPCR of Arabidopsis root hairs). Otherwise there is little basis for the conclusion that depolarization is directly mediated by AUX1 (an alternative explanation for some of the findings is that the 'unknown' receptor is extracellular and interacts primarily with protonated IAA and that at least some of the depolarization and H⁺ fluxes are associated with Ca²⁺-dependent alkalinization; Monshausen et al 2011).

(iii) Several findings in this ms have been published previously. This is sometimes not clearly stated in the text or the relevant papers are cited in a different context, which can create a misleading impression of novelty.

- auxin-induced membrane depolarization in root hairs (Felle and Hepler, 1997, Plant Physiol 114: 39-45);

- pH dependence of depolarization (Felle et al 1991);

- pH-dependent AUX1-mediated auxin uptake (Yang et al, 2006)

- auxin-induced Ca²⁺ signaling in roots (Monshausen et al 2011; Shih et al, 2015, Curr Biol 25:3119-25).

(iv) Some results contradict published literature, but no explanation or discussion is provided:

- In this paper, internalized auxin triggers Ca²⁺ signaling in root hairs, but in root epidermal cells, auxin does not need to be taken up to trigger Ca²⁺ transients (2,4-D elicits Ca²⁺ increase in aux1 mutant; Shih et al, 2015, Curr Biol 25:3119-25).

- In this paper, 1-NAA elicits greater AUX1-dependent membrane depolarization than 2,4-D (Figure S2), but it was previously shown that 1-NAA is not a substrate for AUX1, whereas 2,4-D is (Yang et al 2006; Yang and Murphy, 2009, Plant J 59:179–191).

(v) The conclusion that "AUX1 regulated local auxin signalling in root hairs triggers Ca²⁺ waves that controls auxin action over both short and long-distances" (p.7) is not well supported. It seems entirely possible that iontophoretically injected auxin moves symplastically (and even apoplastically once exported from the cytoplasm) to different parts of the root to trigger DII Venus degradation. The authors could distinguish between those mechanisms by blocking Ca²⁺ signaling (pharmacologically with La³⁺ or maybe genetically in the cngc14 mutant background; Shih et al, 2015, Curr Biol 25:3119-25).

(vi) Little attempt has been made to quantify the concentrations of auxin used to treat root hairs:

- apoplastic application of auxin is via a micropipette (20µm opening) positioned ca 150 µm from the root hair. If the pipette solution contains e.g. 10 µM IAA, a 1s pressure pulse will likely result in a lower auxin concentration at the root hair surface. Given that the authors perform dose-response experiments and calculate half-maximal response concentrations, there should be more effort to determine/calibrate root hair surface concentrations (e.g. co-ejection with a fluorescent dye to calculate dilution factor).

- Intracellular application is performed via iontophoretic injection of a 1nA current over 60 or 300s.

The major charge carrier in the micropipette appears to be IAA at 6.66 mM. The authors should make

some effort to calculate how much auxin is actually introduced. The iontophoretic flux (moles), M , is typically approximated as:

$$M = n \times I \text{ (C/s)} \times T \text{ (s)} / z \times F \text{ (C)},$$

Under the described experimental conditions [$T = 300 \text{ s}$, $I = 1 \text{ nA}$ (or 1 C/s), it seems that a very large amount of auxin is injected into a cell of less than 10 picoliter cytosolic volume. How physiological are these conditions?

(vii) I was surprised that injection of a large (1 nA) current (injection of negative charge) over a period of several minutes did not result in strong hyperpolarization of the membrane. How can this be explained?

Reviewer #3 (Remarks to the Author):

This paper uses a combination of electrophysiology and calcium concentration imaging to investigate the action of AUX1, a membrane protein that considerable evidence indicates mediates the uptake of the hormone auxin. The work is performed primarily in root hairs. The measurements include membrane potential, proton flux, and intracellular calcium concentration.

I have been waiting for one of the findings this paper reports for a very long time. Figures 1 and 2 show that the membrane depolarization triggered by extracellular auxin depends largely on AUX1 and is associated with an influx of protons, not the activation of anion channels as the same research group had previously proposed [Nature (1992) 353: 758 - 762] but do not here discuss. The experimentation appears sound and the results are clear. This result is important because it connects a membrane potential phenomenon known for decades to the AUX1 transporter, thus giving it some functional relevance, but more so because the proton flux data add more evidence to the view that AUX1 functions as a proton-coupled auxin symporter. Although this new data is significant it is not the voltage-clamp evidence needed to establish the proton-auxin symport function for AUX1 with biophysical rigor.

I believe readers will not find the remainder of the paper to be important for the following reasons.

1) Suppression of the depolarization by phosphate has a separate, and I believe probable, explanation not considered by the authors. Root hairs are expected to have high levels of proton-phosphate symport activity. In the presence of phosphate, the conductance of the plasma membrane would be higher than in the absence of phosphate due to the operation of this system. Against this higher background conductance (lower electrical resistance, R), the same amount of auxin-dependent current (i) would necessarily cause a smaller change in membrane voltage ($V=iR$). The auxin transport activity may not be different at different phosphate concentrations. Instead, the effect of this auxin current on the membrane potential is affected by the parallel operation of a phosphate uptake system. The fact that resting membrane potential is phosphate-independent could be due to compensatory increase in proton pumping. I would expect this. The fundamental point is that membrane potential cannot be directly dissected into currents (transport activities). That is why the patch clamp technique, or other voltage-clamp variants so powerful.

2) An inhibitor is used to test for the involvement of the TIR1 auxin receptor. It is expected in the field to test this point with Arabidopsis mutants. I believe Monshausen's cited work on the connection between TIR1 and rapid ionic responses to auxin gives stronger evidence against the conclusion that TIR1 is required

3) Monshausen's group demonstrated a few years ago that auxin triggers a rise in cytoplasmic calcium in root hairs. The large portion of the paper dealing with this phenomenon contains mostly descriptive data. The results do not connect this calcium response to any auxin action.

4) No biological relevance seems conceivable for auxin uptake by root hairs because its concentration in soil must be much lower. Even highly exuded primary metabolites including amino acids do not typically reach micromolar concentrations in soils where microbial turnover is high.

Line 33 – Auxin import is only uphill for the deprotonated anion. Presumably, AUX1 is a proton-auxin anion symporter because the neutral form of auxin which will be by far the most abundant form in any reasonable soil (pH 6-8).

Below the key experiments are described that were conducted and thereafter the comments of the reviewers are addressed point for point:

1. All three reviewers took the specificity of auxinol for inhibition of the TIR1 receptor into question and requested further studies with TIR1/AFB receptor mutants. We therefore compared auxin evoked root hair depolarizations, H⁺ and Ca²⁺-fluxes, of *tir1* and *tir1afb2afb3* loss-of-function mutants, with wild type (Fig. 4 and Fig. S4E, F and G). This revealed that simultaneous loss of TIR1, AFB2 and AFB3 leads to a strong reduction in all three auxin responses, whereas only slight modifications were found upon loss of TIR1 alone. These data thus are in support of an important role of TIR1-like receptors in fast, membrane associated, responses to auxin, as previously concluded, based on the results with auxinole.
2. In the previous version of our manuscript, it was shown that phosphate starvation leads to an enhanced auxin-dependent depolarization. We therefore suggested that low phosphate nutrition accelerates auxin transport in roots. However, reviewer 3 takes the impact of phosphate on IAA-transport into question and offers an alternative explanation for the enhanced depolarization at low phosphate nutrition. For this reason, we measured auxin-induced H⁺-fluxes into roots of phosphate-starved and control seedlings. We found that the H⁺-uptake was increased during the same period after application of IAA, as in which the depolarization was enhanced (Fig. 2C, Fig. S3C). These data are in support of enhanced IAA-uptake in phosphate starved roots.
3. Our experiments show that auxin triggers Ca²⁺ signals in root hairs, which propagate transversal, as well as longitudinal towards the root tip. Reviewer 2 requests further exploration of this response, using La³⁺ as a broad range inhibitor of Ca²⁺-permeable channels and the *cncg14* mutant that is impaired in auxin signalling (Shih et al., 2015). We found that application of 128 μM La³⁺ inhibited auxin-induced depolarizations (Fig. S6A) and the decay of DII-VENUS signals (Fig. 6D), after application of IAA into a single root hair cell. These results support a role for Ca²⁺-

signals in long distance IAA-dependent signalling. Moreover, loss of CNGC14 strongly impaired the depolarization of root hairs evoked by external and internal IAA application, as well as the IAA-dependent uptake of Ca^{2+} (Fig. 5E and F, Fig. S5B, C and D). Apparently, the putative Cyclic-Nucleotide Gated Channel 14 has an essential role fast, membrane associated, responses to auxin.

Reviewer 1:

Comment: I find that the discovery that TIR1 is involved in very rapid signaling processes opens a new concept as to how protein degradation could be involved in processes that take few seconds to activate.

Answer: Thank you, this is a key finding of our manuscript.

Comment 1: To really confirm the role of TIR1 in mediating Ca waves, it would be necessary to examine a KO TIR1 mutant. Pharmacological treatment always rise the concern that they might affect other proteins in a non-specific manner. Thus, it cannot be ruled out that auxinole could be affecting components of the auxin signaling pathway other than TIR1.

Answer: This point of concern was raised by all three reviewers, but we would like to point out that auxinole was designed as a specific antagonist of TIR1 and other members of the auxin-related F box proteins (Hayashi et al., ACS Chem Biol., 2012). The inhibitor was shown to block SCF^{TIR1} -dependent signalling, including the TIR1-AUX/IAA interaction, gravitropism, primary root inhibition, lateral root promotion and root hair formation. Because of its ability to block TIR1-based signalling, auxinole has been used in more than twenty further studies, which confirmed to its ability to block auxin signalling. Although we are not aware of any unspecific side effects of auxinole, we agree with the reviewers that it is worthwhile to back up the data with this inhibitor, with loss of function mutants of TIR1-AFB receptors.

A triple mutants lacking TIR1, AFB2 and AFB3 was compared with wild type and found to be impaired in the auxin-induced root hair depolarization, as well as in H^+ and Ca^{2+} -fluxes (Fig. 4, Fig. 4E, F and G). These results thus support the auxinole data and show that $\text{SCF}^{\text{TIR1/AFB}}$ receptors are essential for fast, membrane-associated, auxin responses.

Comment 2: The results reported on the effect of Pi on auxin transport into root hairs are not completely clear. If Pi signaling leads to an up-regulation of AUX1 expression, a higher level of this auxin transport is expected. Thus, it is not a very novel finding, unless it is proven that Pi signaling affect AUX1 transport efficiency of proton availability in a manner that is independent on the level of AUX1.

Answer: Based on experiments with AUX1p:AUX1-YFP reporter plants, it was recently shown that under low Pi conditions the abundance of AUX1 in the plasma membrane of root epidermal cells increases (Kumar et al., J.Exp.Bot., 2015). We now show that a higher AUX1 abundance correlates with higher auxin transport capacity. This may seem

trivial at first sight, but AUX1 may be post-transcriptionally regulated and a higher amount of AUX1 proteins therefore does not necessarily result in higher transport rates for IAA.

The increased IAA-transport provides an explanation for the impact of phosphate nutrition on root system architecture (RSA). For this reason, the relation of phosphate availability and auxin transport is of major importance to understand how roots optimize nutrient acquisition in poor soils.

Comment 3: It is a bit contradictory that the depolarization experiments reported in this work, show that IAA has a strong preference for IAA than synthetic auxins, while in previous works it has been reported that NAA and 2,4 D have been a higher affinity for auxin uptake transporters than IAA, and that is why they are more active and often used in tissue culture. The authors should explain this discrepancy.

Answer: In the past contradictory results have been published with respect to the efficiency by which IAA and its synthetic analogues trigger auxin responses. In our study, we have focussed on the uptake of auxins by AUX1 in roots. The data of these very early auxin response may differ from that of later responses, in which the affinity of the SCF^{TIR1/AFB} receptors may be essential. Please note that our data match those of Felle et al. (1991), who measured early electrical responses in maize coleoptiles. We now explain our findings in more detail and cite the paper of Felle et al. (1991) in the respective paragraph.

Reviewer 2:

Comment 1: A key conclusion of this work is that TIR1/AFB auxin receptors are required to activate Ca²⁺ channels upon a rise in cytosolic auxin concentration. The discovery of a non-transcriptional signaling role for TIR1/AFB would be a major breakthrough. However, as far as I can tell, the authors deduce this solely on the strength of a pharmacological assay using the TIR1/AFB inhibitor auxinole. I don't understand how the authors can be quite so certain that auxinole is specific to TIR1/AFB? If there is another, not yet identified auxin receptor, this receptor could clearly also be a target of auxinole. It was previously shown that auxin-induced, Ca²⁺-dependent ion signaling also occurs in the tir1 afb2 afb3 triple mutant (Monshausen et al 2011). How can this be reconciled with the conclusions described in this ms? Do root hairs and epidermal cells of the root elongation zone employ very different strategies to activate Ca²⁺ signaling? The authors need to provide better support for such a novel role of TIR1/AFB (e.g. mutant analyses, analysis of auxinole effects in apical root tissues).

Answer: This issue was also raised by reviewer 1 (comment 1) and 3 (comment 2). As suggested by reviewer 2 we conducted experiments with the tir1, afb2 and afb3 triple mutant. This mutant was also used by Monshausen et al. (2011) to study pH changes at the surface of roots, which were evoked by global application of 1 μm IAA. The tir1/afb2/afb3 triple mutant still showed pH changes at the root surface in response to IAA application (Monshausen et al., 2011, Fig. S6), but unfortunately a quantitative comparison with wild type has not been published.

We also conducted experiments with *tir1afb2afb3* seedlings and found that loss of these three auxin receptors impaired the root hair depolarization, as well as the auxin evoked influx of H^+ and Ca^{2+} (Fig. 4, Fig. S4E, F and G). Please note that the triple mutant still showed a residual response to IAA, suggesting a function of additional TIR1-AFB receptors in roots of *Arabidopsis*. This explains why Monshausen et al. (2011) still found a response in the *tir1afb2afb3* and concluded that the TIR-AFB receptors are not involved in rapid auxin responses of roots. They may have gotten to a different conclusion by quantification of their data. Our results with the *tir1afb2afb3* mutant are in line with those obtained with auxinole and suggest an important role for $SCF^{TIR1/AFB}$ receptors in fast, membrane associated, auxin signalling.

Comment 2: It has previously been shown that AUX1 is not normally detectable in root hairs (Jones et al, 2009, Nat Cell Biol 11: 78–84). Unless I missed it, no evidence is provided in this or the accompanying manuscripts that phosphate starvation promotes AUX1 expression in root hairs. The authors must verify that such an increase in AUX1 levels occurs (e.g. using transgenic Arabidopsis expressing AUX:: AUX1-YFP - Jones et al 2009, or via qPCR of Arabidopsis root hairs). Otherwise there is little basis for the conclusion that depolarization is directly mediated by AUX1 (an alternative explanation for some of the findings is that the ‘unknown’ receptor is extracellular and interacts primarily with protonated IAA and that at least some of the depolarization and H^+ fluxes are associated with Ca^{2+} -dependent alkalinization; Monshausen et al 2011).

Answer: Unfortunately, the present literature is inconsistent with respect to expression of *AUX1* in root hairs. Jones et al. (2009) reported that AUX1-YFP proteins cannot be detected in root hair cells, if the expression of AUX-YFP is controlled by the AUX1 promotor. In contrast, other papers with transcriptome- and proteome data indicate that *AUX1* is expressed and present at the protein level in root hairs (e.g. Birnbaum et al., Science 2003; Lan et al. Genome Biology 2013).

We compared root hair cells with non-differentiated epidermal cells and found that the auxin-induced depolarization is slightly larger in non-differentiated cells as in root hairs (Fig. S3A). These results are thus in line with a higher expression of AUX1 in non-differentiated epidermis cells, compared to root hairs. Nevertheless, AUX1-dependent auxin uptake causes a strong depolarization in both cell types, which is to be expected, since root hairs are electrically coupled to the neighbouring root epidermal cells (Wang et al., 2015). Despite of the slightly smaller depolarization in root hair cells, we choose this cell type for our studies, since in this cell type microelectrodes always end up in the cytosol, provided they are impaled at the root hair tip. In epidermal root cells, most of the electrodes also penetrate the vacuolar membrane (Wang et al., 2015), which complicates the membrane potential recordings. Moreover, the cytosolic location enabled us to stimulate cells by current injection of charged molecules, as we did with IAA and Lucifer yellow.

Comment 3: Several findings in this ms have been published previously. This is sometimes not clearly stated in the text or the relevant papers are cited in a different context, which can create a misleading impression of novelty.

- auxin-induced membrane depolarization in root hairs (Felle and Hepler, 1997, Plant Physiol 114: 39-45);
- pH dependence of depolarization (Felle et al 1991);

- pH-dependent AUX1-mediated auxin uptake (Yang et al, 2006)
- auxin-induced Ca^{2+} signaling in roots (Monshausen et al 2011; Shih et al, 2015, *Curr Biol* 25:3119-25).

Answer: We referred to these papers in our manuscript, but because of space limitations we did not discuss them in detail. In the new version of the manuscript we improved this and point out known properties of fast auxin signalling in further detail.

Comment 4: Some results contradict Several published literature, but no explanation or discussion is provided:

a) In this paper, internalized auxin triggers Ca^{2+} signaling in roots hairs, but in root epidermal cells, auxin does not need to be taken up to trigger Ca^{2+} transients (2,4-D elicits Ca^{2+} increase in *aux1* mutant; Shih et al, 2015, *Curr Biol* 25:3119-25)

Answer: We respectfully disagree, since two alternative explanations exist for the observation of Shih et al.:

1. 2,4-D binds to an ‘unknown’ extracellular receptor that triggers Ca^{2+} signals in roots, via yet uncharacterized pathway. This explanation put forward by Shih et al. (2015).
2. 2,4-D is transported into root epidermal cells via an uncharacterized (non-AUX1) transporter in a non-electrogenic manner. This is in line with our observation that 2,4-D causes only a small depolarization of the membrane potential (Fig. S2B; Felle et al., 1991), whereas it is taken up by plant cells (Delbarre et al., 1996). We therefore favour the latter explanation.

b) In this paper, 1-NAA elicits greater AUX1-dependent membrane depolarization than 2,4-D (Figure S2), but it was previously shown that 1-NAA is not a substrate for AUX1, whereas 2,4-D is (Yang et al 2006; Yang and Murphy, 2009, *Plant J* 59:179–191).

Answer: Yang et al. (2006) and Yang and Murphy (2009) expressed AUX1 in *Xenopus* Oocytes and *Schizosaccharomyces pombe*, respectively. In both heterologous expression systems, the AUX1 transport activity was determined by measuring the uptake of ^3H -IAA. The authors show that extracellular application of excess IAA and 2,4-D inhibit uptake of ^3H -IAA, but 1-NAA has no effect. This may either mean that 2,4-D is an alternative substrate of AUX1, or an inhibitor. Our data suggest that AUX1 does not transport 2,4-D and thus it may inhibit ^3H -IAA uptake by AUX1.

Please note that 1-NAA does not elicit AUX1-dependent membrane depolarizations, as suggested by reviewer 2. The depolarization triggered by 1-NAA does not significantly differ between wild type and the *wav5-33* mutant. This points out that 1-NAA depolarizes root hairs in a manner that is independent of AUX1.

Comment 5: The conclusion that “AUX1 regulated local auxin signalling in root hairs triggers Ca^{2+} waves that controls auxin action over both short and long-distances” (p.7) is not well supported. It seems entirely possible that iontophoretically injected auxin moves symplastically (and even apoplastically once exported from the cytoplasm) to different parts of the root to trigger DII Venus degradation. The authors could distinguish between those mechanisms by blocking Ca^{2+} signaling (pharmacologically with La^{3+} or

maybe genetically in the *cngc14* mutant background; Shih et al, 2015, Curr Biol 25:3119-25).

Answer: We agree and disagree with the reviewer. Indeed, the transversal speed of the Ca^{2+} wave, observed following auxin injection is in the order of 5mm/h, which is close to reported transport speed of auxin in roots (Kramer et al., 2011, Trends in Plant Science 16, 461-463). However, the speed of the tip-wards going Ca^{2+} wave is approximately ten times higher and thus it cannot be explained by diffusion. Nevertheless, we further examined the role of Ca^{2+} as suggested by the reviewer, using La^{3+} (a broad range blocker of Ca^{2+} -permeable channels) and the *cngc14* loss-of-function mutant (CNGC14 encodes a putative Ca^{2+} channel that is important for auxin signalling, as shown by Shih et al., 2015). We found that La^{3+} inhibited the auxin-induced depolarization of root hairs cells (Fig. S6A) and reduced the magnitude by which local application of IAA caused a decay of the DII-VENUS signals (Fig. 6D). These results thus support a role for Ca^{2+} -signals in long distance, IAA-dependent, signalling. Loss of CNGC14 even has a more pronounced impact on auxin signalling. The depolarization evoked by external and internal IAA application was strongly impaired in *cngc14*, as well as the IAA-triggered uptake of Ca^{2+} (Fig. 5E and F, Fig. S5B, C and D). Apparently, the putative Cyclic-Nucleotide Gated Channel 14 has an essential role fast, membrane associated, responses to auxin.

Comment 6: Little attempt has been made to quantify the concentrations of auxin used to treat root hairs:

a) apoplastic application of auxin is via a micropipette (20um opening) positioned ca 150 um from the root hair. If the pipette solution contains e.g. 10 uM IAA, a 1s pressure pulse will likely result in a lower auxin concentration at the root hair surface. Given that the authors perform dose-response experiments and calculate half-maximal response concentrations, there should be more effort to determine/calibrate root hair surface concentrations (e.g. co-ejection with a fluorescent dye to calculate dilution factor).

Answer: The reviewer is correct, to remark that the concentrations of auxin at the root surface, will differ from those in application pipettes. We now estimated the decrease in concentration based on experiments with fluorescent dyes. At a distance of 150 μm the dye concentration four-fold diluted in comparison to that in the application pipette. We now added this information to the material and methods.

b) Intracellular application is performed via iontophoretic injection of a 1nA current over 60 or 300s. The major charge carrier in the micropipette appears to be IAA at 6.66 mM. The authors should make some effort to calculate how much auxin is actually introduced. The iontophoretic flux (moles), M , is typically approximated as: $M = n \times I \text{ (C/s)} \times T \text{ (s)} / z \times F \text{ (C)}$, Under the described experimental conditions [$T = 300 \text{ s}$, $I = 1\text{nA}$ (or 1 C/s), it seems that a very large amount of auxin is injected into a cell of less than 10 picoliter cytosolic volume. How physiological are these conditions?

Answer: If a current of -1 nA is applied through a micro-electrode, it will provoke the movement of anions into the cytosol, but at the same time also that of cations from the cytosol into microelectrode. The concentration of K^+ in the cytosol of root hairs will exceed that of IAA in the electrode and both K^+ and H^+ have a much higher mobility than

IAA. The fluxes of K⁺ and H⁺ thus will exceed those of Lucifer Yellow and IAA. A further complication are local charges at the electrode tip, which may cause difference in electrode resistance for anion and cations. For this reason, it is extremely difficult to calculate the cytosolic IAA concentration that was evoked by current injection and we step back from making estimates.

Comment 7: I was surprised that injection of a large (1 nA) current (injection of negative charge) over a period of several minutes did not result in strong hyperpolarization of the membrane. How can this be explained?

Answer: In the paper of Wang et al. (2015) we describe that micro electrodes in the cytosol of root epidermis cells record a conductance of approximately 100 nS. Injection of -1 nA therefore will cause a hyperpolarization of -10 mV. We found very similar values for bulging root hair cells (Wang et al., 2015, Fig. 3). The low resistance measured by electrodes in the cytosol is probably due to electrical connections between root cells via plasmodesmata. Please note that we also show a hyperpolarization of approximately -10 mV upon injection of -1 nA in Fig. S5A and B.

Reviewer 3:

Comment 1: Suppression of the depolarization by phosphate has a separate, and I believe probable, explanation not considered by the authors. Root hairs are expected to have high levels of proton-phosphate symport activity. In the presence of phosphate, the conductance of the plasma membrane would be higher than in the absence of phosphate due to the operation of this system. Against this higher background conductance (lower electrical resistance, R), the same amount of auxin-dependent current (i) would necessarily cause a smaller change in membrane voltage ($V=iR$). The auxin transport activity may not be different at different phosphate concentrations. Instead, the effect of this auxin current on the membrane potential is affected by the parallel operation of a phosphate uptake system. The fact that resting membrane potential is phosphate-independent could be due to compensatory increase in proton pumping. I would expect this. The fundamental point is that membrane potential cannot be directly dissected into currents (transport activities). That is why the patch clamp technique, or other voltage-clamp variants so powerful.

We agree with the reviewer that voltage-clamp recordings are superior to studies in which only the membrane potential is monitored. However, in intact seedlings, root hairs are electrically coupled to neighbouring cells and thus the plasma membrane conductance of these cells cannot be studied with voltage clamp technique. Instead, we used scanning ion-selective electrodes, to estimate ion fluxes at the root surface and link these fluxes to membrane potential changes.

We used the scanning H⁺-selective electrode to study the impact of phosphate nutrition on IAA-induced H⁺-fluxes (Fig. S3C). These experiments revealed that phosphate starvation stimulates H⁺-uptake in a short period following IAA application. This period overlaps with that in which the auxin-induced depolarization is affected by phosphate nutrition (Fig. 2C). These results thus strongly suggest that phosphate starvation

stimulates IAA-uptake, which in turn leads to an enhanced depolarization of root hair cells.

Comment 2: An inhibitor is used to test for the involvement of the TIR1 auxin receptor. It is expected in the field to test this point with Arabidopsis mutants. I believe Monshausen's cited work on the connection between TIR1 and rapid ionic responses to auxin gives stronger evidence against the conclusion that TIR1 is required

Answer: All three reviewers requested us to back up the results with auxinole with mutants that lack TIR1-like receptors (Reviewer 1 and 2, comment 1). We have now conducted experiments with the *tir1afb2afb3* triple mutant as well as with the *tir1* single mutant. We found that loss of TIR1, AFB2 and AFB3 clearly impaired the auxin-induced depolarization of root hairs, as well as H^+ and Ca^{2+} uptake, but these responses were only slight altered in the *tir1* mutant (Fig. 4, Fig. S4E, F and G). These data thus strongly support our results with auxinole and suggest an important role for $SCF^{TIR1/AFB}$ receptors in fast, membrane associated, auxin signalling.

Please note that the study of Monshausen et al. (2011) showed that surface pH changes still occurred in the *tir1afb2afb3* mutant (Monshausen et al. (2011), Fig. S6), but they did not quantify the difference between the triple mutant and wild type.

Comment 3: Monshausen's group demonstrated a few years ago that auxin triggers a rise in cytoplasmic calcium in root hairs. The large portion of the paper dealing with this phenomenon contains mostly descriptive data. The results do not connect this calcium response to any auxin action.

Answer: The reviewer is correct to mention that Monshausen's group reported auxin-induced calcium signaling in root hairs, just as Felle and Hepler already reported on the influence of auxin on the cytosolic Ca^{2+} gradient of root hairs in 1997. However, in the manuscript we (i) link the Ca^{2+} response in root hairs to the depolarization triggered by IAA, (ii) compare Ca^{2+} responses elicited by several different auxins, (iii) show that $SCF^{TIR1/AFB}$ receptors are important of auxin-induced Ca^{2+} signalling and (iv) demonstrate that the putative CNGC14 channel is required for the auxin-dependent influx of Ca^{2+} . These new data will be very valuable to discover the molecular mechanism by which auxin evokes cytosolic Ca^{2+} signals. Moreover, our experiments with the DII-VENUS reporter indicate that Ca^{2+} signals are important for long distance auxin signalling in roots.

Comment 4: No biological relevance seems conceivable for auxin uptake by root hairs because its concentration in soil must be much lower. Even highly exuded primary metabolites including amino acids do not typically reach micromolar concentrations in soils where microbial turnover is high.

Answer: Root hair cells probably do not take up IAA from the soil, but instead from neighbouring cells, just as non-differential cells in the root epidermis. This kind of auxin transport is also supported by the two manuscripts that were co-submitted with ours,

which show that AUX1 is important to facilitate shoot-ward auxin transport from the root apex to differentiation zone, in order to promote hair elongation.

Please note that we used bulging root hair cells of which most of the cell surface is still in contact with other cells (see Fig. 5A). We choose to impale the tip of bulging root hairs cells to ensure that the microelectrodes end up in the cytosol, whereas they penetrate the vacuolar membrane in the majority of the non-differentiated epidermis cells (Wang et al., 2015).

Comment 5: Line 33 – Auxin import is only uphill for the deprotonated anion. Presumably, AUX1 is a proton-auxin anion symporter because the neutral form of auxin which will be by far the most abundant form in any reasonable soil (pH 6-8).

Answer: Just as explained above we do not expect the root hair cells to take up auxin from the soil und natural conditions. Instead we suppose that auxin is derived from cell wall of neighbouring cells. The pH of the root apoplast ranges from 5 – 5.5 and thus the proportion of IAA^- : IAAH is about 60 : 40 (pH 5) or 85 : 15 (pH 5.5). Given the very negative membrane potential of root hair cells and a pH difference of about 2 units root hairs may accumulate auxin to concentrations that exceed that of the apoplast by a factor of 10^5 .

Reviewers' comments:

Reviewer #2 (Remarks to the Author):

In this revised manuscript, Dindas et al confirm that loss of TIR1, AFB2 and AFB3 activity mimics the inhibitory effects of auxinole treatment. This greatly strengthens their conclusion that the TIR1/AFB receptor complex is involved not just in the well-established transcriptional pathway but also in very rapid auxin-triggered ion signaling. This is genuinely exciting but some of the new results are difficult to interpret.

It appears that the proposed AUX1-dependent auxin-induced depolarization is not only undetectable in auxinole-treated wt and *tir1 abf2 afb3* triple mutants but also in *cngc14* mutants (where no auxin-induced Ca^{2+} transients are found) and after La^{3+} treatment (where auxin surprisingly triggers a sustained Ca^{2+} increase; Figure S6). The authors speculate that "the increased $[Ca^{2+}]_{cyt}$ - as observed in the presence of the Ca^{2+} channel blocker La^{3+} - seem to inhibit AUX1-mediated IAA transport, possibly through activation of Ca^{2+} -dependent protein kinases" (p9, line296). Inhibition of AUX1 activity both by loss of TIR/AFB proteins, resting (*cngc14*) and high (La^{3+}) Ca^{2+} levels is rather surprising, especially given that *cngc14* mutants show no obvious *aux1* mutant phenotypes. The authors should at least confirm that auxin uptake is indeed inhibited. To this end they could perform qPCR to demonstrate that auxin-induced upregulation of e.g. *AUX/IAA19* or *-6* does not occur in *cngc14* mutants/ La^{3+} pre-treated roots. If it is not inhibited, this would suggest that the H^+/IAA^- stoichiometry is changed - how could this be explained?

Another question I raised previously was not really addressed in this manuscript. The authors propose that injection of auxin triggers a local Ca^{2+} increase, which then propagates through the root tip to modulate auxin action and/or transport. They have now tested the effect of the Ca^{2+} channel blocker La^{3+} and found a partial inhibition of DII Venus degradation in the root tip. The problem here is that in the authors' hands, La^{3+} did not block Ca^{2+} changes; in fact, pretreatment with La^{3+} triggered a more sustained Ca^{2+} elevation after auxin treatment.

It remains entirely possible that the authors are seeing the effect of auxin movement from the site of injection. Obviously, injected compounds do diffuse into neighboring cells, as Lucifer Yellow fluorescence is not just detectable in the injected cell but also in cells throughout the tissue shown in Figure 5A (unless that is bleed-through from R-GECO1?). Given that the electrode barrel used for injection contained 0.5 mM LY but 6.6 mM IAA, it must be assumed IAA also moved out of the cell. In any case, it should be possible to estimate the concentration of injected IAA based on LY fluorescence intensity.

If the authors cannot inhibit Ca^{2+} signaling with La^{3+} , it may be possible to 'dip' the tip of a (longer) root hair into the opening of a pipette containing auxin for a few seconds. If the root hair Ca^{2+} signal is sufficient to elicit a wave of Ca^{2+} through the root tissue, it should be observable under these conditions.

Some other points:

- the authors should be more careful about phrasing with regard to applied auxin concentration.

Although it is now stated in the Materials that there is about a four-fold dilution of auxin during pressure ejection, K_m values are apparently still calculated as if no dilution occurs. The dilution factor should be mentioned in the Results section with a note that calculations are based on undiluted concentrations.

- p8, line 250: " La^{3+} inhibited both, IAA triggered Ca^{2+} signaling..."

p10, line 315: "Blocking Ca^{2+} signaling via La^{3+} inhibits DII-VENUS degradation and thus distal auxin signaling..."

The authors have not blocked Ca^{2+} signaling with La^{3+} (see Figure S6).

- It would be nice to include a graph showing the time course of R-GECO1 fluorescence in auxinole- and auxin-treated roots, not just the max fluorescence slope (Figure 3).

Referee #2 still raised a number of points of concern that needed to be addressed before publication. Below we will discuss these comments point for point.

In this revised manuscript, Dindas et al confirm that loss of TIR1, AFB2 and AFB3 activity mimics the inhibitory effects of auxinole treatment. This greatly strengthens their conclusion that the TIR1/AFB receptor complex is involved not just in the well-established transcriptional pathway but also in very rapid auxin-triggered ion signaling. This is genuinely exciting but some of the new results are difficult to interpret.

It appears that the proposed AUX1-dependent auxin-induced depolarization is not only undetectable in auxinole-treated wt and *tir1 abf2 afb3* triple mutants but also in *cngc14* mutants (where no auxin-induced Ca²⁺ transients are found) and after La³⁺ treatment (where auxin surprisingly triggers a sustained Ca²⁺ increase; Figure S6). The authors speculate that “the increased [Ca²⁺]_{cyt} – as observed in the presence of the Ca²⁺ channel blocker La³⁺ – seem to inhibit AUX1-mediated IAA transport, possibly through activation of Ca²⁺-dependent protein kinases” (p9, line296). Inhibition of AUX1 activity both by loss of TIR/AFB proteins, resting (*cngc14*) and high (La³⁺) Ca²⁺ levels is rather surprising, especially given that *cngc14* mutants show no obvious *aux1* mutant phenotypes. The authors should at least confirm that auxin uptake is indeed inhibited. To this end they could perform qPCR to demonstrate that auxin-induced upregulation of e.g. AUX/IAA19 or -6 does not occur in *cngc14* mutants/La³⁺ pre-treated roots. If it is not inhibited, this would suggest that the H⁺/IAA- stoichiometry is changed – how could this be explained?

1. In the previous version manuscript, we suggest that Ca²⁺-signaling affects auxin transport. This hypothesis is questioned by the referee and alternative explanations for our observations are proposed.

We based our hypothesis that cytosolic Ca²⁺ signals modulate auxin transport on the following results: i) In contrast to wild type, auxin fails to induce a depolarization and Ca²⁺ influx in root cells of the putative Ca²⁺ channel mutant *cngc14*. ii) The auxin- dependent depolarization is prevented by La³⁺, a broad range Ca²⁺ channel blocker. The referee poses the following question: *Is the absence of the auxin-induced depolarization due to a lack of auxin transport, or because of a change in stoichiometry of the AUX1 transporter?*

We approached this question with the following experiments:

i) As suggested by the referee we used quantitative PCR to test if IAA-uptake into roots is affected in the *cngc14* mutant, or by pre-incubation with La³⁺. To this purpose seedlings were exposed to 0.1 μM IAA for 1 min and the expression level of IAA19 was determined 1 hour later (Fig. S5E). We observed that IAA19 expression is enhanced more than 3-fold by IAA in wild type seedlings. The auxin-induced IAA19 expression is impaired by La³⁺ but essentially absent in the *cngc14* mutant. The outcome of these experiments is thus well in line with our hypothesis that Ca²⁺ signals modulate IAA transport in roots. Finally, we found that IAA19 expression is controlled by the SCF_{TIR/AFB}

complex, since IAA19 basal expression levels are far below wild type levels in the *tir1/afb2/afb3* receptor mutant and could not be induced by auxin application.

ii) In order to test if the stoichiometry of AUX1 is altered, we performed simultaneous H⁺ and Ca²⁺ flux estimation experiments with the *cngc14* mutant. In contrast to wild type, IAA-induced neither Ca²⁺, nor H⁺ uptake, as is now shown in Fig. 5E. This proves that AUX1-dependent IAA/H⁺ cotransport is impaired in the *cngc14* loss-of-function mutant. Even at a stoichiometry of 1 H⁺ to 1 IAA⁻, which would not have affected the membrane potential (see Fig. 5F) proton fluxes would have been detected with scanning ion selective electrodes.

Another question I raised previously was not really addressed in this manuscript. The authors propose that injection of auxin triggers a local Ca²⁺ increase, which then propagates through the root tip to modulate auxin action and/or transport. They have now tested the effect of the Ca²⁺-channel blocker La³⁺ and found a partial inhibition of DII Venus degradation in the root tip. The problem here is that in the authors' hands, La³⁺ did not block Ca²⁺ changes; in fact, pretreatment with La³⁺ triggered a more sustained Ca²⁺ elevation after auxin treatment.

It remains entirely possible that the authors are seeing the effect of auxin movement from the site of injection. Obviously, injected compounds do diffuse into neighboring cells, as Lucifer Yellow fluorescence is not just detectable in the injected cell but also in cells throughout the tissue shown in Figure 5A (unless that is bleed-through from R-GECO1?). Given that the electrode barrel used for injection contained 0.5 mM LY but 6.6 mM IAA, it must be assumed IAA also moved out of the cell. In any case, it should be possible to estimate the concentration of injected IAA based on LY fluorescence intensity.

If the authors cannot inhibit Ca²⁺ signaling with La³⁺, it may be possible to 'dip' the tip of a (longer) root hair into the opening of a pipette containing auxin for a few seconds. If the root hair Ca²⁺ signal is sufficient to elicit a wave of Ca²⁺ through the root tissue, it should be observable under these conditions.

2. The second point of concern regards the auxin-induced Ca²⁺ signal, which propagates from the root hair cell that is injected with auxin, towards the root tip. In the manuscript, we proposed that this Ca²⁺ wave affects auxin signaling in cells that are at distance from the cell, which was initially stimulated by auxin. The referee takes this hypothesis into question and argues that auxin may just move from the cell that was injected, into adjacent cells. The referee points out that Lucifer Yellow, which is co-injected with auxin, also diffuses to neighboring cells, based on Fig. 5A. It is likely that the referee is misled by the weak fluorescent signal of Yellow Cameleon (YC3.6) in this image, which is expressed in these plants together with R-GECO1 (Keinath et al., 2015). In root hairs, Lucifer Yellow

rapidly accumulates in the vacuole and therefore remains in the cell in which it was injected.

Nevertheless, we agree with the referee is that small solutes injected in to root hairs are likely to move into neighboring cells, either by diffusion via plasmodesmata, or by transport across the plasma membranes. However, the concentration of solutes that are injected into a single cell will rapidly drop with distance, as the solutes will move in three dimensions. IAA injected into a single cell thus could rapidly move into neighboring cells and trigger a response, such as a rise in the cytosolic Ca^{2+} level. However, the velocity at which these responses are induced would quickly decrease with distance.

In contrast to cell to cell movement of solutes, the Ca^{2+} signal triggered by auxin injection into a single epidermal root cell propagated at a constant velocity of 1.56 cm/h towards the root tip (Fig. S6C). In this respect, the propagating Ca^{2+} signal is very similar to the Ca^{2+} waves found in various other organisms, as reported by Jaffe (2010) and Koenigsberger et al. (2010). In the new version of the manuscript we explain our findings in more detail and explain that the propagating Ca^{2+} signal has all hall marks of a travelling Ca^{2+} wave.

Some other points:

- the authors should be more careful about phrasing with regard to applied auxin concentration. Although it is now stated in the Materials that there is about a four-fold dilution of auxin during pressure ejection, Km values are apparently still calculated as if no dilution occurs. The dilution factor should be mentioned in the Results section with a note that calculations are based on undiluted concentrations.

3. In the first revision of the manuscript, we added information about the gradient of the auxin concentration, when it is applied to roots by pressure injection via micro capillaries. This information was added to the material and methods, but the referee requests that this information is also added to the results. We followed the reviewer's suggestion and added this information to first paragraph of the results ('for apparent IAA concentration see Methods), the figure captions of Figs. 1 and 2, as well as Fig. S4.

- p8, line 250: " La^{3+} inhibited both, IAA triggered Ca^{2+} signaling..."
p10, line 315: "Blocking Ca^{2+} signaling via La^{3+} inhibits DII-VENUS degradation and thus distal auxin signaling..."

The authors have not blocked Ca^{2+} signaling with La^{3+} (see Figure S6).

4. We agree with the referee that La^{3+} does not seem to block plasma membrane Ca^{2+} channel, at the concentration of 128 μM . Instead it seems to cause a sustained auxin- induced rise in the cytosolic Ca^{2+} concentration. We therefore changed the text of the results accordingly and no longer write that La^{3+} blocks

Ca²⁺ signaling. However, La³⁺ does interfere with the auxin-induced Ca²⁺ signals and we thus explain that it modulates DII-VENUS degradation and auxin distal auxin signaling.

- It would be nice to include a graph showing the time course of R-GECO1 fluorescence in auxinole- and auxin-treated roots, not just the max fluorescence slope (Figure 3).

The referee requests an additional graph, which shows the time course of R-GECO1 fluorescence intensity changes, induced by auxin, in control as well as in auxinole-treated roots. We have added such a graph to Fig. S4E and further show that local auxin application induces a Ca²⁺ wave that propagates at constant velocity, as explained above (Fig. S6C).

REVIEWERS' COMMENTS:

Reviewer #2 (Remarks to the Author):

In this revised manuscript, the authors provide convincing evidence that CNGC14 not only mediates auxin-dependent Ca²⁺ signaling, but is also required for TIR1/AFB-dependent AUX19 expression. This supports the authors hypothesis that CNGC14 is required for auxin uptake and satisfactorily addresses my previous comment.

I am less persuaded by the authors arguments concerning whether it is an auxin or a Ca²⁺ wave that is propagated through the root tissue after auxin microinjection.

The authors conclude that it must be a Ca²⁺ wave based on the velocity of wave propagation (15 mm per h), as auxin waves would quickly slow down with distance. However, this is not supported by the literature, where velocities of over 15 mm per h are routinely measured over larger distances (basipetal movement of auxin from root tip; polar auxin transport in shoots), mediated by a range of auxin transporters. This is why providing some information on how much auxin was injected is important.

However, given that this is not the major point of the ms, I think it is sufficient if this is (briefly) discussed in the text and does not require additional experimentation.

Reviewer #2 (Remarks to the Author):

In this revised manuscript, the authors provide convincing evidence that CNGC14 not only mediates auxin-dependent Ca²⁺ signaling, but is also required for TIR1/AFB-dependent AUX19 expression. This supports the authors hypothesis that CNGC14 is required for auxin uptake and satisfactorily addresses my previous comment.

I am less persuaded by the authors arguments concerning whether it is an auxin or a Ca²⁺ wave that is propagated through the root tissue after auxin microinjection.

The authors conclude that it must be a Ca²⁺ wave based on the velocity of wave propagation (15 mm per h), as auxin waves would quickly slow down with distance. However, this is not supported by the literature, where velocities of over 15 mm per h are routinely measured over larger distances (basipetal movement of auxin from root tip; polar auxin transport in shoots), mediated by a range of auxin transporters. This is why providing some information on how much auxin was injected is important.

However, given that this is not the major point of the ms, I think it is sufficient if this is (briefly) discussed in the text and does not require additional experimentation.

Answer from the Authors

We understand the concerns of the reviewer.

In order to meet the uncertainty regarding auxin transport out of the injected cell the last paragraph of the results was modified in accordance with the request of reviewer #2. We now explain that that auxin transport can occur at a velocity of 15 mm h⁻¹, but because of the constant velocity of propagation, the recorded Ca²⁺ signals most likely resemble Ca²⁺ waves, which have been observed in a variety of eukaryotes (lines 601 and 602).